**EMBO** *reports*

# Hyperactive PLCG1 induces cell-autonomous and bystander T cell activation and drug resistance

Longhui Zeng [1], Xinyan Zhang[1], Yiwei Xiong[1], Kazuki Sato [1], Nicole Hajicek[2], Yasunori Kogure [3], Keisuke Kataoka[3,4], Seishi Ogawa[5,6,7], John Sondek[2] & Xiaolei Su [1,8,9,10,11]✉

## Abstract

Phospholipase C gamma 1 (*PLCG1*) has been identified as the most frequently mutated gene in adult T-cell leukemia/lymphoma, suggesting a critical function of PLCG1 in driving T cell activation. However, it remains unclear how these mutations regulate T cell physiology and pathology. Here, we investigate three common leukemia/lymphoma-associated mutations (R48W, S345F, and D1165H). We discover that these mutations induce hyperactive T cell signaling and cause pro-survival phenotypes. PLCG1 mutants enhance LAT condensation, calcium influx, and ERK activation. They also promote T cell proliferation, upregulate cell adhesion molecules, induce cell aggregation, and confer resistance to Vorinostat, an FDA-approved drug for cutaneous T-cell lymphoma. The resistance depends on ERK signaling and can be reversed with an ERK inhibitor. Interestingly, PLCG1 mutants also induce bystander drug resistance in nearby cells expressing wild-type PLCG1. Mechanistically, alpha smooth muscle actin, which is specifically induced by PLCG1 mutants, directly binds PLCG1 to promote its activation. These results demonstrate that hyperactive PLCG1 promotes T cell survival and drug resistance by inducing non-canonical signaling.

**Keywords** Actin; Condensation; ERK; PLCG1; T Cell
**Subject Categories** Cancer; Immunology; Signal Transduction

## Introduction

Phospholipase C gamma 1 (PLCG1) protein serves as a key signaling molecule linking the upstream activation of transmembrane receptors (e.g. TCR, EGFR, PDGFR) to the downstream pathways leading to cytoskeleton remodeling, membrane fusion, and transcriptional induction (Chen and Simons, 2021). In T lymphocytes, TCR activation leads to the phosphorylation of LAT,

a transmembrane adaptor protein (Zhang et al, 1998) that forms biomolecular condensation to promote TCR signaling (Huang et al, 2019; Su et al, 2016). Phosphorylated LAT recruits PLCG1 to the plasma membrane, resulting in the hydrolysis of $PIP_2$ to generate $IP_3$ and DAG, which induce calcium influx and MAPK signaling (Balagopalan et al, 2015; Courtney et al, 2018).

PLCG1 is a multi-domain phospholipase that has both enzyme-dependent and independent functions. The high-resolution structure of essentially full-length PLCG1 (Hajicek et al, 2019) revealed the three-dimensional arrangement of PLCG1 domains. The regulatory domain cluster, which is composed of the sPH, nSH2, cSH2, and SH3 domains, interacts with the catalytic core to mask the substrate-binding site. When Y783 is phosphorylated (potentially by Itk (Qi and August, 2007)), it binds the cSH2 domain, triggering a conformational change that leads to the exposure of the catalytic site so that PLCG1 cleaves $PIP_2$ to generate the second messengers $IP_3$ and DAG (Gresset et al, 2010; Hajicek et al, 2019). In addition to the enzyme-dependent function, PLCG1 also has a scaffolding function: the nSH2, cSH2, and SH3 domains interact with LAT and Sos1 to promote condensation of the LAT complex and enhance TCR signal transduction (Wada et al, 2022; Zeng et al, 2021).

The pathological relevance of PLCG1 signaling was revealed by genome profiling of clinical samples. PLCG1 mutations were identified in cutaneous T-cell lymphoma (Patel et al, 2020; Vaque et al, 2014), angioimmunoblastic T-cell lymphoma (Wang et al, 2017), angiosarcoma (Behjati et al, 2014), and immune dysregulation diseases (Tao et al, 2023). Notably, PLCG1 is the most frequently mutated gene in adult T-cell leukemia/lymphoma (ATLL); about 36% of patients acquired mutations in PLCG1 (Kataoka et al, 2015). This suggested PLCG1 as a key signaling molecule for T cell malignancy. The point mutations identified in ATLL spread along the entire PLCG1 with a few high-frequency spots. Namely, R48W sits on an N-terminal PH domain that potentially interacts with plasma membranes (Falasca et al, 1998). S345F is in the catalytic TIM barrel whereas D1165H is located on the C-terminal C2 domain that interacts with the membrane through calcium (Fig. 1A) (Ananthanarayanan et al, 2002; Lomasney et al, 2012). Although it remains unclear how R48W affects the conformation of PLCG1, both S345F and D1165H are

[1]Department of Cell Biology, Yale School of Medicine, New Haven, CT, USA. [2]Department of Pharmacology, The University of North Carolina at Chapel Hill, Chapel Hill, NC, USA. [3]Division of Molecular Oncology, National Cancer Center Research Institute, Tokyo, Japan. [4]Division of Hematology, Department of Medicine, Keio University School of Medicine, Tokyo, Japan. [5]Department of Pathology and Tumor Biology, Graduate School of Medicine, Kyoto University, Kyoto, Japan. [6]Institute for the Advanced Study of Human Biology (WPI-ASHBi), Kyoto University, Kyoto, Japan. [7]Kindai University Faculty of Medicine, Osakasayama, Japan. [8]Yale Cancer Center, New Haven, CT, USA. [9]Yale Center for Immuno-Oncology, New Haven, CT, USA. [10]Yale Center for Systems and Engineering Immunology, New Haven, CT, USA. [11]Yale Stem Cell Center, New Haven, CT, USA.
✉E-mail: Xiaolei.su@yale.edu

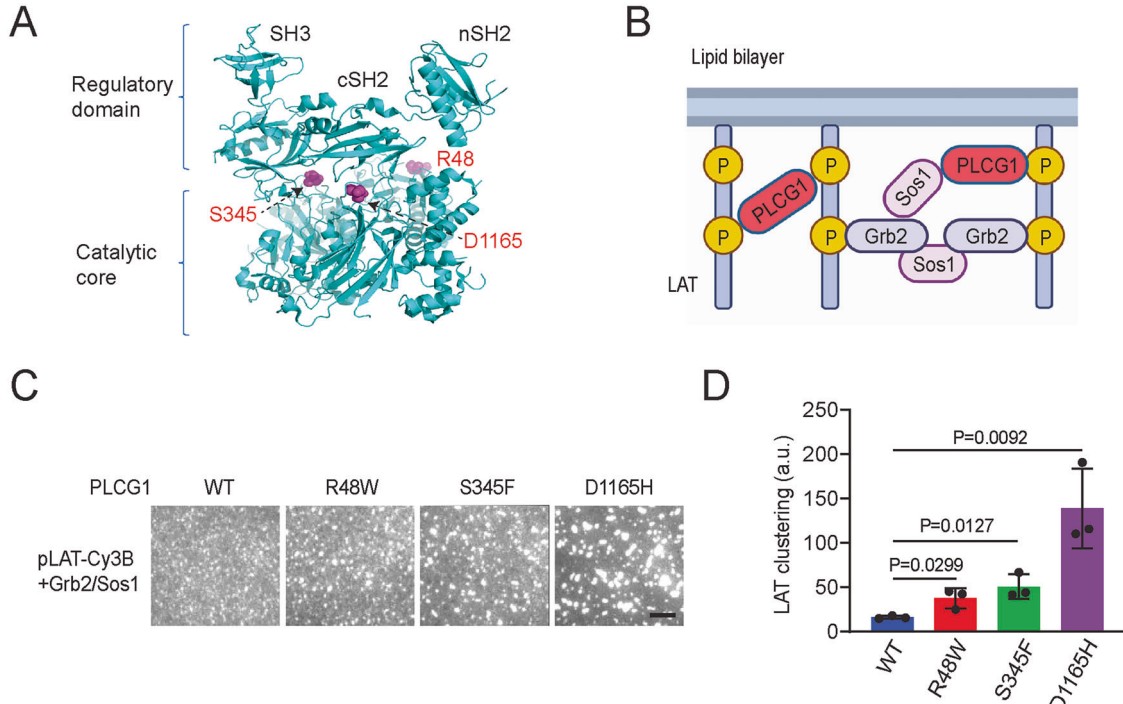

**Figure 1. PLCG1 acquiring ATLL-associated mutations promotes LAT condensation in vitro.**

(A) Location of the three T-cell leukemia/lymphoma-associated mutations in the structure of PLCG1 (PDB: 6PBC). (B) Schematics of biochemical reconstitution of LAT condensation on supported lipid bilayers. (C) TIRF microscopy revealed that PLCG1 mutations enhanced LAT condensation on bilayers at physiological concentrations. Cy3B-labeled LAT at 300 molecules/$\mu m^2$ was incubated with 300 nM Sos1 (the fragment that contains proline-rich motifs), 3000 nM Grb2 and 50 nM PLCG1 for 0.5 h before imaging. Scale bar: 5 $\mu m$. (D) Quantification of LAT clustering as normalized variation. Shown are mean ± SD from $n = 3$ biological replicates. Unpaired two-tailed $t$ test was used. Source data are available online for this figure.

located at the autoinhibitory interface between the regulatory cluster and catalytic core; these mutations are expected to favor an open conformation of PLCG1 (Hajicek et al, 2019). Indeed, biochemical assays showed an enhanced lipase activity of all three mutants although R48W displayed a much milder enhancement than S345F or D1165H (Hajicek et al, 2019). Moreover, these mutants trigger enhanced signaling downstream of PLCG1 including NFAT, AP-1, and NF-kB in cell line models (Patel et al, 2020; Vaque et al, 2014). However, it remains unclear whether hyperactivation of PLCG1 is sufficient to drive T cell proliferation and reduce cell death. The mechanisms by which these mutants contribute to cellular phenotypes linked to tumor progression also remain largely unexplored. Additionally, it remains unknown whether the signaling and phenotypes induced by hyperactive PLCG1 simply mimic those triggered by TCR activation, or if novel functions of PLCG1 may arise outside of the TCR signaling network.

Therefore, we decided to investigate the mechanism and cellular consequence of hyperactive PLCG1 signaling using these ATLL-related mutants. We found that PLCG1 mutations enhanced LAT condensation, calcium influx and ERK phosphorylation. They promoted the activation and proliferation of human primary T cells. Using the T cell lymphoma line Hut78 as a model, we found that PLCG1 mutants induced cell aggregation by enhancing ICAM-1 expression. ICAM-1 engaged with integrin LFA-1 to increase ERK activation. The hyperactive ERK rendered cell resistance to vorinostat, an FDA-approved drug for cutaneous T-cell lymphoma

and this resistance to vorinostat can be reversed by an ERK inhibitor. This resistance was also observed in cells expressing the wild-type PLCG1 when they were co-clustered with PLCG1 mutant cells, suggesting a bystander drug resistance induced by PLCG1 mutants. To determine if hyperactive PLCG1 induces signaling beyond canonical TCR network, we performed gene expression profiling and found genes in smooth muscle contraction were specifically induced by PLCG1 mutants but not TCR-stimulated wild-type samples. Alpha smooth muscle actin, which is highly expressed in PLCG1 mutants, directly bound PLCG1 and promoted the activation of PLCG1. Together, our work reveals how hyperactive PLCG1 affects T cell signaling and drug resistance and uncovers insights on the pathogenesis mechanism of T-cell leukemia and lymphoma. Our work also highlights the neo-signaling induced by hyperactive PLCG1 mutants beyond traditional TCR signaling network.

## Results

### PLCG1 acquiring ATLL-associated mutations promote LAT condensation in vitro

Our previous work demonstrated that PLCG1 promotes LAT condensation and enhances signaling beyond its lipase function in generating second messengers $IP_3$ and DAG. This is achieved

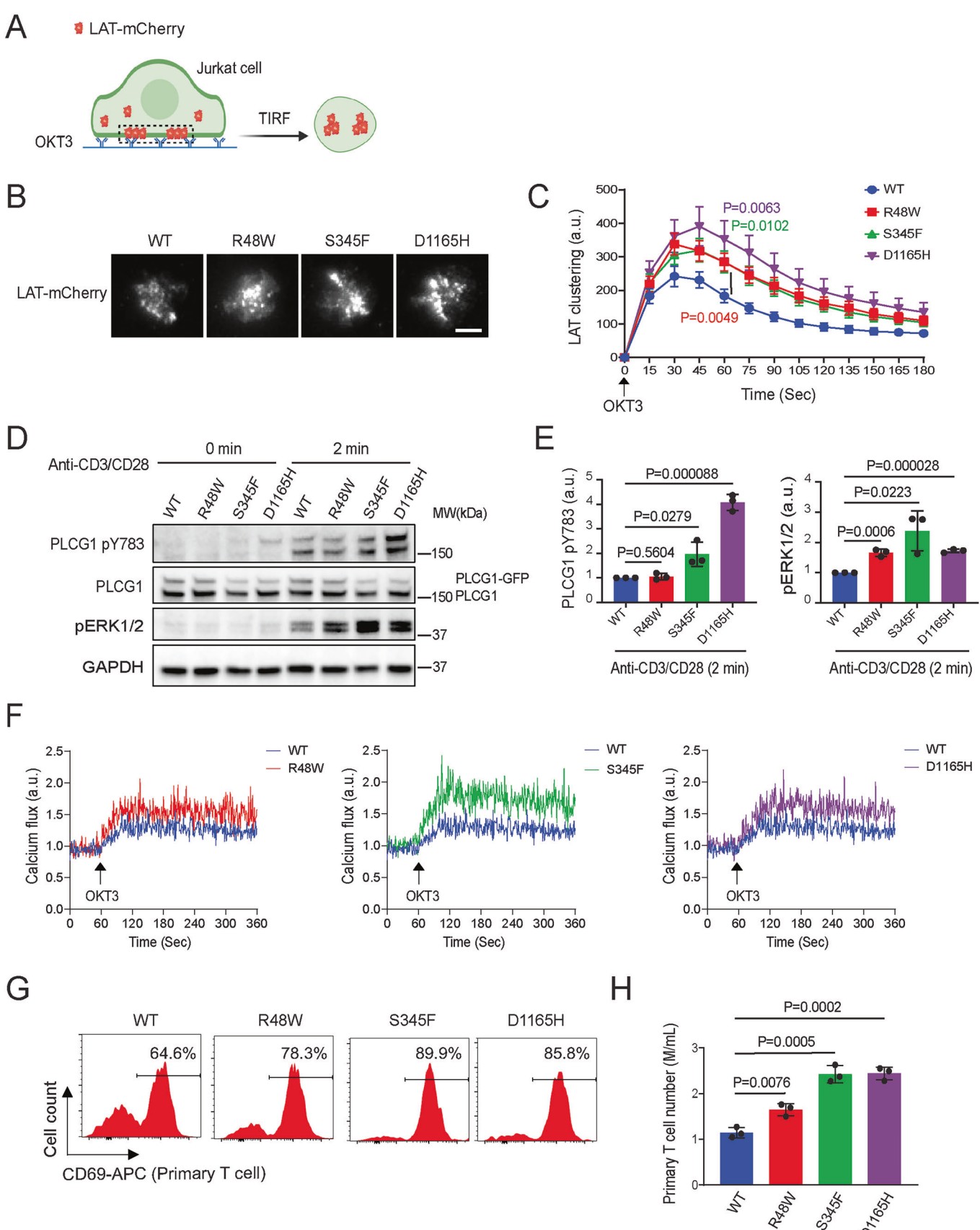

**Figure 2. PLCG1 mutants increase TCR-triggered T cell activation.**

(A) Schematics of imaging T cell activation in live cells. Jurkat T cells expressing LAT-mCherry and PLCG1 WT or mutants were plated and activated on OKT3 antibody-coated glass. The formation of LAT condensates on plasma membranes were monitored by time-lapsed TIRF microscopy. (B) Representative images of Jurkat T cells stimulated with glass-coated OKT3. Images shown were 45 s after cell landing. Scale bar: 5 μm. (C) Quantification of LAT clustering during T cell activation. Shown are mean ± SEM. $N = 24$–30 cells. Unpaired two-tailed test was used for mutation groups compared to WT under stimulation at 60 s. R48W vs WT: $P = 0.0049$, S345F vs WT: $P = 0.0102$, D1165H vs WT: $P = 0.0063$. (D) Immunoblot analysis of Jurkat T cells harboring PLCG1 WT or indicated mutated variant stimulated with anti-CD3 and anti-CD28 antibodies. (E) Quantification of (D). Shown are mean ± SD from $n = 3$ biological replicates. Unpaired two-tailed $t$ test was used. (F) Calcium flux monitored by flow cytometry following TCR activation. Jurkat T cells expressing calcium sensor GCaMP6s and PLCG1 WT or mutation were stimulated by OKT3 antibody and monitored by flow cytometry in a continuous recording mode. (G) Activation of human primary T cells expressing PLCG1 WT or mutants. The expression of CD69 was determined by flow cytometry 14 days after T cells were infected with lentivirus encoding PLCG1 WT or mutants. (H) Proliferation of human primary T cells expressing PLCG1 WT or mutants. The cell number was quantified 14 days after T cells were infected with lentivirus encoding PLCG1 WT or mutants. Shown are mean ± SD from $n = 3$ biological replicates. Unpaired two-tailed $t$ test was used. Source data are available online for this figure.

through crosslinking LAT and binding partner Sos1 via the SH2 and SH3 domains on PLCG1 (Zeng et al, 2021). To determine how ATLL mutations of PLCG1 affect LAT condensation, we implemented a supported lipid bilayer-based reconstitution assay that we developed before (Su et al, 2017; Zeng and Su, 2023). Briefly, the cytoplasmic domain of LAT was purified and phosphorylated by ZAP70 at four key tyrosine sites (Su et al, 2017), labeled with maleimide-Cy3B on a C-terminal cysteine, and attached to a Ni-NTA functionalized supported lipid bilayer through an N-terminal polyhistidine tag. The wild-type and three mutants of PLCG1 (R48W, S345F, and D1165H), together with Grb2 and Sos1 (the fragment that contains proline-rich motifs) were purified as described before (Hajicek et al, 2019; Zeng et al, 2021) (Fig. EV1A) and added in solution. Total internal reflection fluorescence (TIRF) microscopy was implemented to monitor LAT condensation with PLCG1, Grb2, and Sos1 (Fig. 1B) under physiologically relevant concentrations (Zeng et al, 2021). Interestingly, all three PLCG1 mutants induced higher condensation of LAT as compared to the wild-type PLCG1 (2-fold for R48W, 3-fold for S345F, and 9-fold for D1165H) (Fig. 1C,D). This can be potentially explained by the open conformation of PLCG1 mutants which renders more accessibility of their SH2 domain to the phosphotyrosines on LAT (Hajicek et al, 2019). To determine how PLCG1 mutants affect the liquidity of LAT condensates, we performed fluorescence recovery after photobleaching (FRAP) analysis on LAT condensates. No significant difference was found between the wild-type and mutants (Fig. EV1B), suggesting the liquid-like property of LAT condensates remained similar between the wild-type and mutants. Together, data from biochemical reconstitution suggested that PLCG1 mutants directly enhanced LAT condensation.

## ATLL-associated PLCG1 mutants increase TCR-triggered T cell activation

To determine if the ATLL mutations of PLCG1 promote LAT condensation in T cells, we constructed Jurkat T cell lines stably expressing LAT-mCherry and wild-type or mutant PLCG1 through lentiviral transduction. We kept the endogenous copy of PLCG1 in these cells because these mutations are heterozygous in patients. The Jurkat cells were dropped onto glass-coated with OKT3, an anti-CD3 epsilon antibody that activates the TCR signaling. TIRF microscopy revealed the formation of LAT condensates as Jurkat cells spread on the imaging glass (Fig. 2A). We found that, consistent with results from the above biochemical reconstitution assay, PLCG1 mutants

enhanced LAT condensation in activated Jurkats (1.5-fold for R48W and S345F, 1.9-fold for D1165H) (Fig. 2B,C). In addition to the scaffolding function in promoting LAT condensation, two of the PLCG1 mutants S345F and D1165H increased phosphorylation of PLCG1, a marker for the activation of PLCG1, by 2-fold and 4-fold, respectively (Fig. 2D,E). These results motivated us to further determine LAT and PLCG1 downstream signaling. We found that Jurkat cells expressing the three mutants showed a 1.7–2.4-fold of ERK phosphorylation (Fig. 2D,E) and a 1.4–1.9-fold of calcium influx (Fig. 2F) as compared to those expressing the wild-type PLCG1.

To determine how PLCG1 mutants affect primary T cell activation, we primed human T cells isolated from healthy donors using dynabeads, and infected these cells with lentivirus encoding the wild-type or three mutants of PLCG1. T cells expressing the mutants displayed a 1.2–1.4-fold of CD69 expression (Fig. 2G), a T cell activation marker, and a 1.4- to 2.1-fold of proliferation (Fig. 2H), as compared to the wild-type PLCG1. These results were repeated using T cells from a different donor (Fig. EV1C,D). Together, these data showed that PLCG1 mutants promoted TCR signaling transduction and T cell activation in both the Jurkat T cells line and human primary T cells.

## Hyperactive PLCG1 is sufficient to trigger T cell activation and cytokine production without TCR engagement in the T cell lymphoma line Hut78

To determine how PLCG1 mutants affect signaling in the context of T-cell leukemia and lymphoma, we used Hut78, a human T lymphoma cell line, as a model. Hut78 displayed a signaling signature more similar to that of activated primary T cells than Jurkat cells (Bartelt et al, 2009). We infected Hut78 with lentivirus encoding the wild-type or mutant PLCG1. To determine whether these mutations are sufficient to trigger activation without TCR engagement, we performed the following measurements in the absence of TCR stimuli. We found that even though the expression levels of the ectopically introduced mutants were lower than those of the ectopically introduced wild-type PLCG1, an increase in phospho-PLCG1 (pY783, a marker for PLCG1 activation) was observed in S345F (1.9-fold) and D1165H (9-fold) (Fig. 3A,B). We also found a higher phosphorylation of ERK (1.8–2.3-fold) in all three mutants and a higher expression of Bcl2 (3–3.3-fold), an anti-apoptosis protein (Fig. 3A,B). It is noted that the expression of these ectopically expressed PLCG1 mutants (they contained a GFP tag and migrated more slowly on the gel than the endogenous PLCG1) was much higher than that of the endogenous PLCG1.

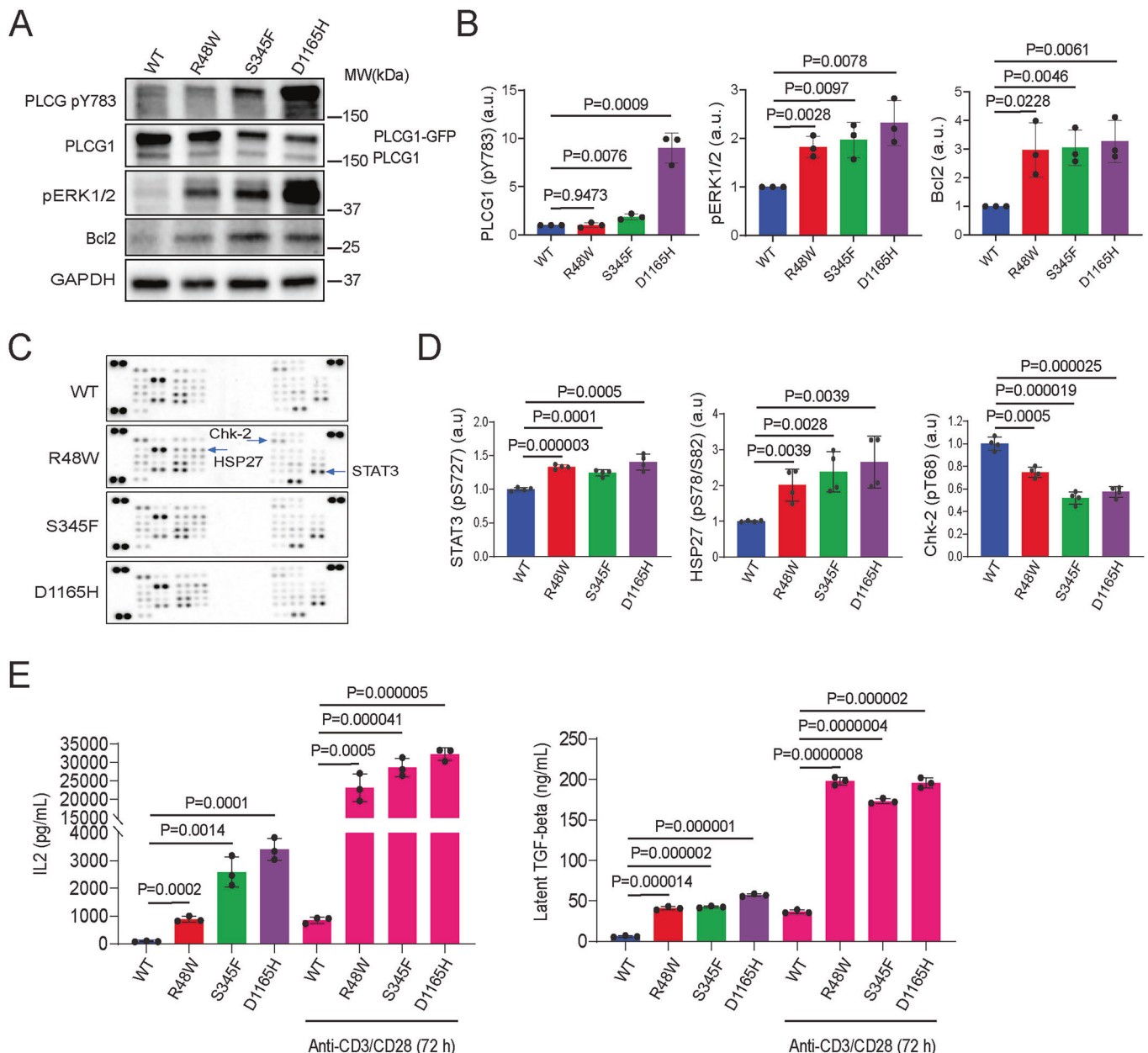

**Figure 3. Hyperactive PLCG1 is sufficient to trigger T cell activation and cytokine production without TCR engagement.**

(A) Immunoblot analysis of signaling in Hut78 cells stably expressing PLCG1 WT or mutants (without TCR activation). (B) Quantification of (A). Shown are mean ± SD from *n* = 3 biological replicates. Unpaired two-tailed *t* test was used. (C) Profiling of protein kinase phosphorylation. Hut78 cells expressing PLCG1 WT or mutations were lysed, and applied to the proteome profiler human phospho-kinase array kit for analysis. The three kinases with altered phosphorylation were highlighted by blue arrows. (D) Quantification of the phosphorylation level of kinases. Shown are mean ± SD from *n* = 4 biological replicates. Unpaired two-tailed *t* test was used. (E) ELISA analysis of IL2 and TGF-beta secretion by Hut78 cells in resting versus activation states. Hut78 cells were activated by anti-CD3/CD28 antibodies for 72 h. Shown are mean ± SD from *n* = 3 biological replicates. Unpaired two-tailed t-test was used. Source data are available online for this figure.

Therefore, we re-infected Hut78 with lentivirus with a lower titer so that the ectopically expressed PLCG1 was at a similar level to the endogenous one (Fig. EV1E,F). In this case, we still observed enhanced phospho-PLCG1 (2–12.1-fold), phospho-ERK (1.4–1.6-fold), and Bcl2 (2.1–2.9-fold) in PLCG1 mutants as compared to the wild-type PLCG1 (Fig. EV1E,F).

To systematically determine the intracellular signaling triggered by the PLCG1 mutants, we profiled the key kinase phosphorylation using

a human phospho-kinase array kit. We identified a few kinases whose phosphorylation was consistently altered in all three mutants (Fig. 3C). These include the enhanced phosphorylation of STAT3 (1.2–1.4-fold), which promotes effector T cell differentiation, enhanced phosphorylation of HSP27 (2–2.6-fold), which has an anti-apoptosis function, and reduced phosphorylation of Chk-2 (1.4–2-fold), a tumor suppressor (Fig. 3D). These signaling outcomes are consistent with the notion that PLCG1 mutants promote cell growth and inhibit cell death.

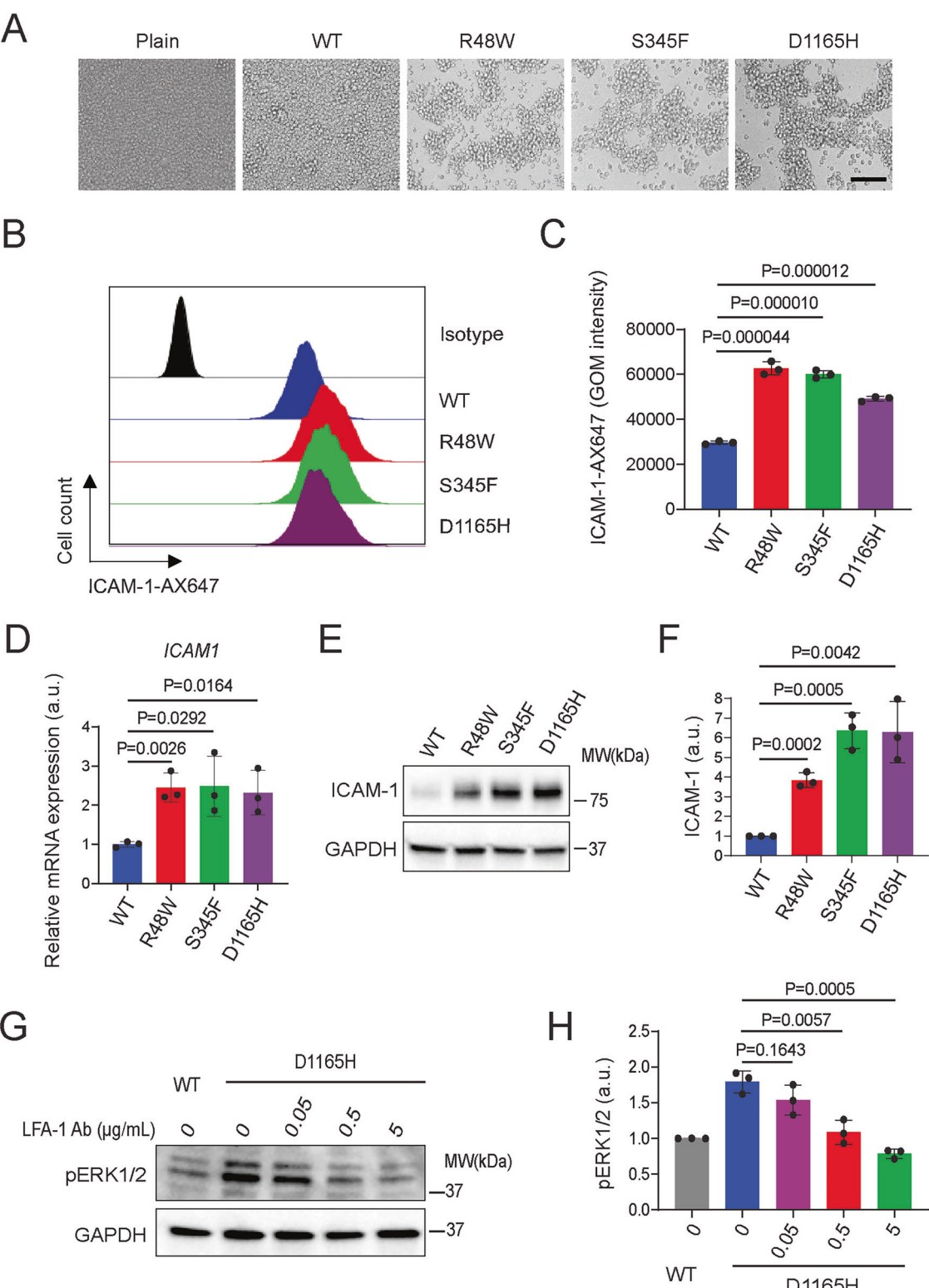

Cytokine production is one of the key functional consequences of TCR pathway activation. Therefore, we determined the production of IL-2 and TGF-beta, the two signature cytokines in the TCR pathway, in PLCG1 mutants. We found a higher level of IL-2 and TGF-beta were released by cells expressing the PLCG1 mutants both in resting

(IL-2: 10-fold for R48W, 32-fold for S345F, and 42-fold for D1165H; TGF-beta: 7-fold for R48W, 7-fold for S345F, and 9-fold for D1165H) and stimulated statuses (IL-2: 27-fold for R48W, 34-fold for S345F, and 38-fold for D1165H; TGF-beta: 5-fold for all three mutants) (Fig. 3E). Together, the above data suggested that PLCG1 mutants are

◄ **Figure 4. Hyperactive PLCG1 signaling induces aggregation of Hut78.**

(A) Aggregation of Hut78 cells expressing PLCG1 mutants. Scale bar: 100 μm. (B) Flow cytometry revealed ICAM-1 expression on Hut78 expressing PLCG1 WT or mutants. (C) Quantification of (B). GOM is geometric mean fluorescence intensity. Shown are mean ± SD from $n = 3$ biological replicates. Unpaired two-tailed *t* test was used. (D) The ICAM-1 mRNA level as determined by qPCR. Shown are mean ± SD from $n = 3$ biological replicates. Unpaired two-tailed *t* test was used. (E) The ICAM-1 protein level as determined by western blot. (F) Quantification of (E). Shown are mean ± SD from $n = 3$ biological replicates. Unpaired two-tailed *t* test was used. (G) Blocking ICAM-1 and LFA-1 interaction inhibited ERK activation. LFA-1 blocking antibody was added to Hut78 expressing PLCG1 D1165H for 36 h. (H) Quantification of (D). Shown are mean ± SD from $n = 3$ biological replicates. Source data are available online for this figure.

sufficient to drive hyperactive PLCG1 signaling; they trigger PLCG1 downstream pathways even in the absence of upstream TCR engagement in Hut78 cells.

## Hyperactive PLCG1 signaling induces aggregation of Hut78 cells

During the culture of Hut78 cells, we found that cells expressing the PLCG1 mutants, but not the wild-type, formed cell-cell aggregates (Fig. 4A). To determine if this aggregation is mediated by soluble factors secreted or by cell adhesion molecules, we plated plain Hut78 cells in conditioned media obtained from cells expressing the mutant PLCG1. We found that the conditioned media did not induce cell aggregation (Fig. EV2A), arguing against a mechanism that the cell aggregation was solely mediated by soluble factors. On the other hand, we determined the expression of common adhesion molecules and found a significant increase of ICAM-1 at the mRNA (2.3–2.5-fold), whole-cell protein expression (3.7–6.4-fold), and cell surface protein expression (1.7–2.1-fold) levels in Hut78 expressing PLCG1 mutants (Fig. 4B–F). ICAM-1 is the ligand for integrin LFA-1 on T cell surface. LFA-1 was expressed at a comparable level between cells expressing the wild-type and mutant PLCG1 (Fig. EV2B). To determine if this ICAM-1-LFA-1 interaction contributes to downstream signaling, we treated Hut78 with antibodies blocking ICAM-1-LFA-1 interactions. Indeed, we find a dose-dependent reduction in ERK phosphorylation (Fig. 4G,H) (LFA-1 was shown to activate ERK in T cells (Cassioli et al, 2021; Sharma et al, 2018)). Together, these data suggested that PLCG1 mutants induced cell-cell aggregation in conjugation with ICAM-1-LFA-1 signaling to enhance ERK activation.

## Hyperactive PLCG1 confers cell-autonomous and bystander resistance to HDAC inhibitors

The aforementioned data showed that ATLL-related PLCG1 mutations increased ERK phosphorylation. Previous work showed that KRAS-induced ERK signaling conferred cell resistance to vorinostat, a histone deacetylase (HDAC) inhibitor and FDA-approved drug for treating cutaneous T-cell lymphoma (Wang et al, 2016). Therefore, we determined the sensitivity of Hut78 to HDAC inhibitors. We found that PLCG1 mutations enhanced cell viability after treatment with vorinostat (2.7–3-fold) (Fig. 5A); consistently, cells expressing PLCG1 mutants were stained with a lower level of apoptosis marker annexin-V (2.7–4.7-fold) (Fig. 5B,C). Similar resistance was revealed using other HDAC inhibitors belinostat (1.6–2.1-fold) and panobinostat (1.7–2.2-fold) (Fig. EV3A,B).

Because mutations in tumors are usually heterogenous among individual cells, next we determined if PLCG1 mutants cause

bystander resistance in adjacent cells that do not acquire mutations. We mixed plain Hut78 cells with Hut78 cells expressing either wild-type or D1165H PLCG1 tagged with GFP in a 1:1 ratio and treated the cell mixture with vorinostat. We found that the plain Hut78 cells mixed with cells expressing D1165H showed lower apoptosis (2.7-fold) as compared to the control group that were mixed with cells expressing wild-type PLCG1 (Fig. EV3D,E). This suggested a bystander resistance induced by PLCG1 mutants. What is the mechanism underlying this bystander resistance? We reasoned that PLCG1 mutants induced the expression of ICAM-1, which can bind LFA-1 on both the plain Hut78 or Hut78 expressing PLCG1 mutants and induce co-aggregation. Indeed, when mixing plain Hut78 (labeled with a far-red dye) with Hut78 expressing either the wild-type or D1165H (labeled with GFP), we found that plain Hut78 cells co-aggregated with cells expressing PLCG1 D1165H (Fig. EV2C,D), suggesting the bystander resistance is mediated through cell-cell interactions. LFA1 activation was reported to increase ICAM1 expression and ERK activation (Owaki et al, 2008; Sharma et al, 2018). Indeed, we found that ICAM1 surface expression was 1.5-fold and ERK phosphorylation was 1.9-fold in bystander Hut78 cells co-cultured with Hut78 cells expressing PLCG1 D1165H as compared to those co-cultured with PLCG1 WT (Fig. EV2E–H).

To explore the mechanism of resistance to HDAC inhibitor, we decided to inhibit ERK and STAT3, which we showed were upregulated in PLCG1 mutants. We found that Hut78 cells expressing D1165H showed a similar cell viability to the wild-type PLCG1 in the presence of an ERK inhibitor LY3214996 whereas LY3214996 itself did not change cell viability (Fig. 5D). In contrast, the STAT3 inhibitor napabucasin did not affect the drug resistance of D1165H (Fig. EV3C). This suggested that ERK signaling mediates PLCG1 mutants-induced resistance to HDAC inhibitors; inhibiting ERK can reverse the drug resistance to vorinostat.

## Hyperactive PLCG1 signaling induces a distinct gene profile from TCR activation

PLCG1 sits in one of the many signaling branches that are triggered downstream of TCR. To determine if the PLCG1 mutants only trigger a branch of TCR signaling or they induce pathways beyond the TCR signaling network, we performed bulk RNA sequencing on Hut78 expressing the wild-type or mutant PLCG1. We also included a group in which cells expressing wild-type PLCG1 were activated by anti-CD3/CD28 antibodies as a conventional way to fully activate TCR. We discovered a total of 227 genes that are upregulated or downregulated in all three mutants (Fig. 6A; Dataset EV1). Among the 227 genes, only 64 genes were shared between the mutants and TCR activation group; the rest 163 genes are uniquely

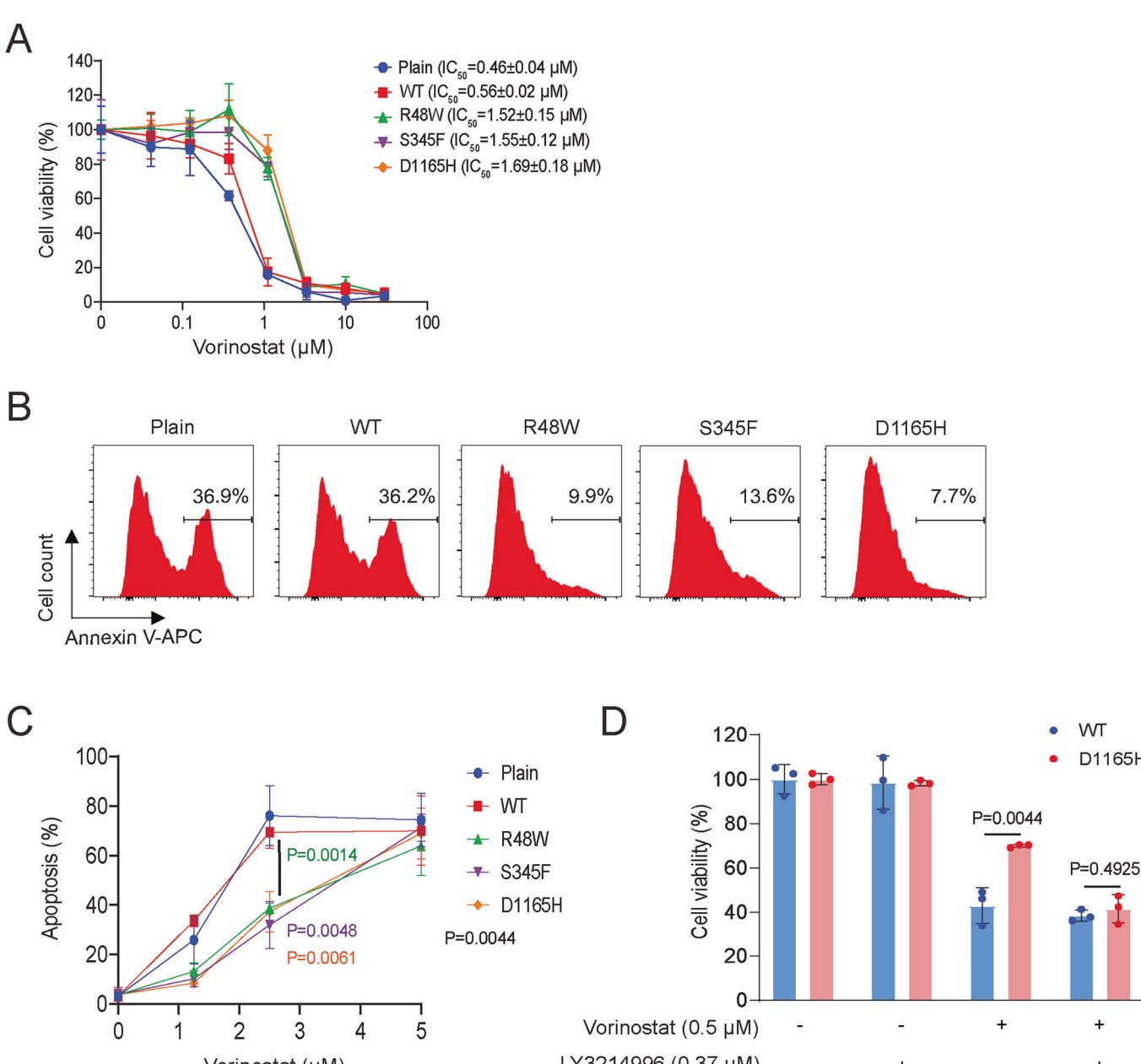

**Figure 5. Hyperactive PLCG1 confers Hut78 resistance to HDAC inhibitors.**

(A) PLCG1 mutations conferred Hut78 resistance to vorinostat. The plain group is Hut78 cells without ectopically expressing PLCG1. The CCK8 assay was used to detect viable cell number after vorinostat treatment for 72 h. Shown are mean ± SD from n = 3 biological replicates. (B) PLCG1 mutations decreased vorinostat-induced apoptosis. Hut78 cells were incubated with 1.25 μM vorinostat for 60 h. (C) Quantification of vorinostat-induced apoptosis. Shown are mean ± SD from n = 3 biological replicates. The unpaired two-tailed t test was used to compare WT to R48W (P = 0.0014), S345F (P = 0.0048), D1165H (P = 0.0061) under 2.5 μM vorinostat treatment. (D) ERK inhibition mitigated vorinostat resistance in Hut78 cells. ERK inhibitor LY3214996 at noncytotoxic concentration 0.37 μM abolished the resistance to vorinostat in Hut78 expressing PLCG1 D1165H. Shown are mean ± SD from n = 3 biological replicates. Unpaired two-tailed t test was used. Source data are available online for this figure.

altered in hyperactive PLCG1 group but not the TCR activation one (Fig. 6B; Dataset EV1). We performed pathway analysis on these 163 genes and found pathways including these related to macrophage activation and cytokine storm were altered (Fig. 6C). Among these, the top upregulated pathway relates to smooth muscle contraction, which contains genes including integrins (e.g. *ITGB5*, *ITGA1*), smooth muscle actin gamma (*ACTG2*), and alpha

smooth muscle actin (alpha-SMA, *ACTA2*) (Dataset EV1). We confirmed, by flow cytometry, that PLCG1 mutations caused an increase in the cell surface expression of *ITGB5* (Integrin beta 5, 1.6–3.3-fold), *ITGA1* (CD49a, 2–2.4-fold), and *SLAMF1* (CD150, 1.6–3.1-fold) (Fig. EV4B). Because alpha-SMA is the key marker of smooth muscle differentiation pathway, we further confirmed the high expression of alpha-SMA (4.2-fold for R48W, 7.9-fold for

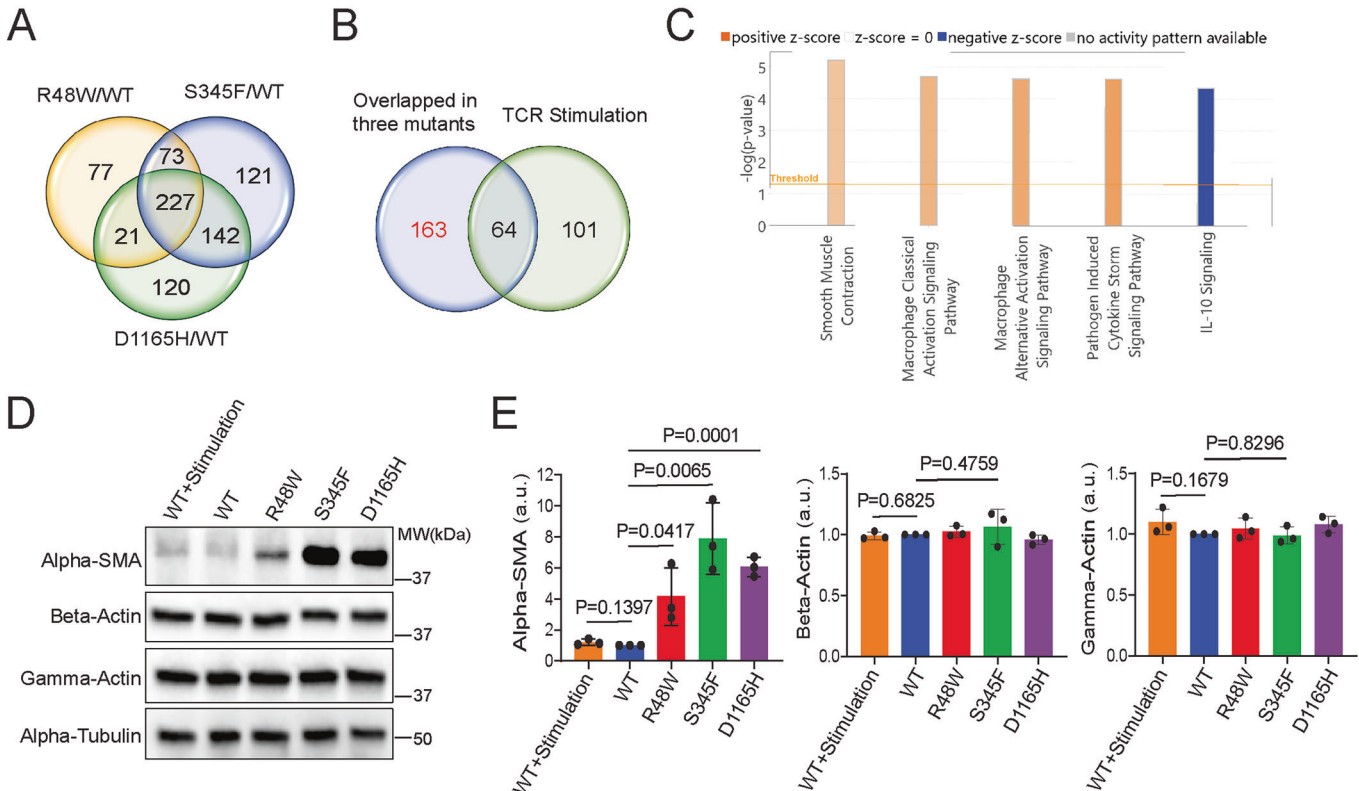

**Figure 6.  Hyperactive PLCG1 signaling induces a distinct gene profile from TCR activation.**

(A) Venn plot illustrates the gene expression difference in the R48W, S345F and D1165H group as compared to the WT PLCG1. RNA-seq analysis was used to profile gene expression in Hut78 cells expressing WT or mutant PLCG1. Threshold: Log2 fold change >1 and FDR < 0.001 when comparing to the WT group. (B) Comparing gene expression profile between hyperactive PLCG1 signaling and TCR signaling. In the TCR group, Hut78 expressing PLCG1 WT was activated by anti-CD3/CD28 antibodies for 3 days. (C) Qiagen ingenuity pathway analysis (IPA) revealed pathways enriched in the gene set uniquely triggering by hyperactive PLCG1 signaling but not TCR signaling. (D) Immunoblot analysis of actin isoform expression in Hut78 cells. WT+Stimulation: PLCG1 WT was stimulated by anti-CD3/CD28 antibodies for 2 days. (E) Quantification of the actin expression. Shown are mean ± SD from $n = 3$ biological replicates. Unpaired two-tailed $t$ test was used. Source data are available online for this figure.

S345F, and 6.1-fold for D1165H) in Hut78 expressing PLCG1 mutants by western blot (Fig. 6D). In contrast, the expressions of universally present beta actin and gamma actin were similar between the wild-type and mutant PLCG1 (Fig. 6E). Together, these results showed hyperactive PLCG1 signaling caused distinct gene expression profile from TCR activation.

## Alpha smooth muscle actin-dependent activation of PLCG1 mutants

To determine the regulators of the hyperactive PLCG1 signaling in ATLL mutants, we performed pull-down assays to identify PLCG1-interacting partners. Hut78 cells expressing the GFP-tagged wild-type or mutant PLCG1 were lysed. PLCG1 and binding partners were isolated using anti-GFP antibody-coated beads and profiled by SDS-PAGE. We found a few specific bands appeared in the S345F and D1165H but not the WT samples (Fig. 7A). These bands were cut out and sent for mass spectrometry analysis. They were identified as myosin heavy chain (around 220 kDa), myosin light chain (around 20 kDa) and actin (around 40 kDa) (Dataset EV2). The actin pulled down included a mixture of beta actin, gamma actin and alpha smooth muscle actin (SMA). This is consistent with

previous reports showing all three isoforms of actin co-polymerize into filaments (Drew and Murphy, 1997). Because alpha-SMA but not beta or gamma actin was upregulated in cells expressing PLCG1 D1165H (Fig. 6D,E), we reasoned that alpha-SMA could be a key factor in mediating binding to PLCG1. Indeed, the recombinant PLCG1 D1165H protein binds directly to filamentous alpha-SMA protein in an actin co-pelleting assay (Fig. 7B,C). It is noted that the wild-type PLCG1 can also directly bind alpha-SMA, suggesting the binding to alpha-SMA is not restricted to mutant PLCG1.

To determine whether alpha-SMA affects PLCG1 activation in cells, we knocked out the gene encoding alpha-SMA (ACTA2) in Hut78 cells, which resulted in a 1.6-fold reduction in PLCG1 phosphorylation in the D1165H group, which had a hyperphosphorylation in PLCG1 when alpha-SMA is present (Fig. 7D,E). Complementary to the genetic approach, we performed pharmacological perturbation of actin polymerization in Hut78 using drugs that either depolymerize (latrunculin) or stabilize actin (jasplakinolide). We found a 1.4-fold decrease in PLCG1 activation with latrunculin and a 1.4-fold increase in activation with jasplakinolide in cells expressing PLCG1 D1165H (Fig. 7F,G). Together, these data suggested that the alpha smooth muscle actin binds and stimulates the activation of PLCG1.

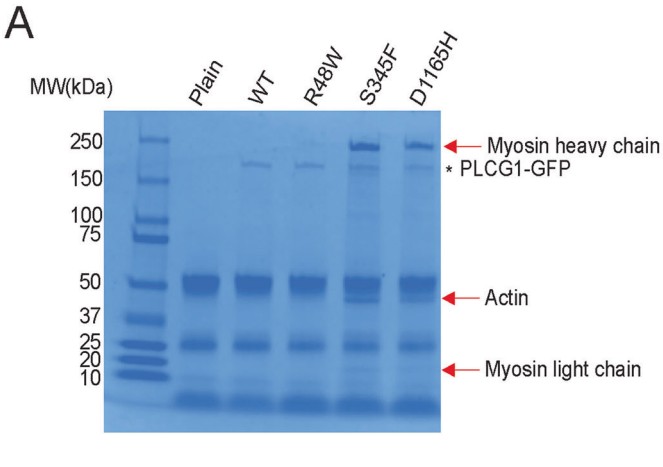

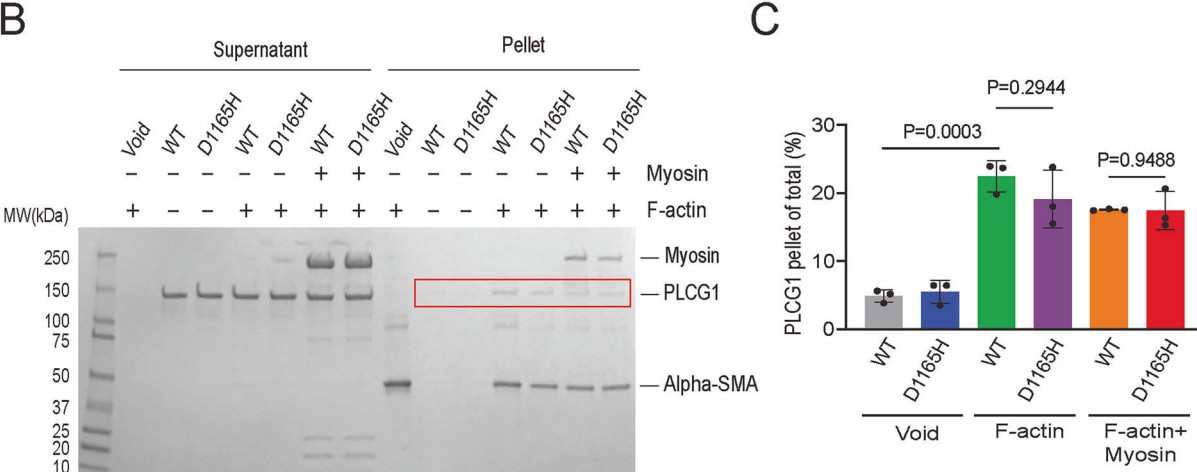

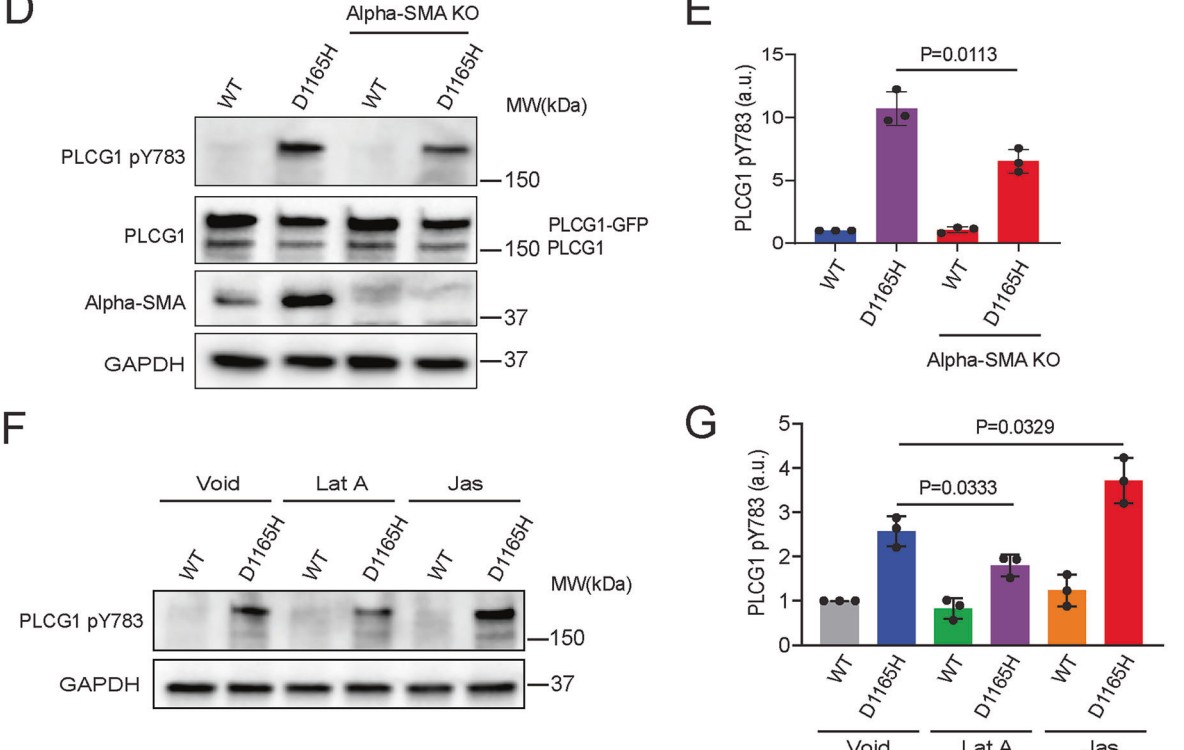

**Figure 7. Alpha smooth muscle actin-dependent activation of PLCG1 bearing ATLL mutations.**

(A) Immunoprecipitation assay to identify PLCG1-binding partners. Hut78 cells expressing PLCG1-GFP (WT or mutant) were lysed, pulled down by beads coated with protein G and an anti-GFP antibody-coated, and applied for SDS-PAGE. The specific bands stained by Coomassie Blue were cut and identified using mass spectrometry. The plain sample is Hut78 cells without exogenously expressed PLCG1. (B) Pelleting assay to determine the direct interaction between PLCG1 and filamentous actin in vitro. Input: 4 µM α-SMA (pre-assembled into filaments), 0.4 µM PLCG1, and 1 µM myosin. (C) Quantification of PLCG1 binding to F-actin. Shown are mean ± SD from $n = 3$ biological replicates. Unpaired two-tailed $t$ test was used. (D) Immunoblotting of PLCG1 phosphorylation in alpha-SMA knockout Hut78 cells. (E) Quantification of PLCG1 phosphorylation from (D). Shown are mean ± SD from $n = 3$ biological replicates. Unpaired two-tailed t-test was used. (F) Regulation of PLCG1 phosphorylation by filamentous actin in Hut78 cells. Actin filaments were destabilized and stabilized with the treatment of 0.5 µM latrunculin A (Lat A) or 0.15 µM jalapinolate (Jas), respectively for 0.5 h. (G) Quantification of PLCG1 phosphorylation. Shown are mean ± SD from $n = 3$ biological replicates. Unpaired two-tailed-t test was used. Source data are available online for this figure.

## Hyperactive PLCG1 signaling in human primary T cells and ATLL patient samples

We further validated the canonical and noncanonical signaling mechanisms of ATLL-associated PLCG1 mutations using human primary T cells. Primary T cells were isolated from PBMCs obtained from healthy donors, activated using Dynabeads, and transduced with lentivirus encoding either wild-type (WT) or mutant PLCG1. Consistent with observations in Hut78 cells, PLCG1 mutations promoted canonical ERK phosphorylation (1.6–2.1-fold) and non-canonical upregulation of alpha-SMA expression (2.7–2.8-fold) (Fig. 8A). Flow cytometry analysis showed that PLCG1 mutations increased the surface expression of ICAM-1 (1.5–1.8-fold) (Fig. 8B,C). These findings prompted us to examine gene expression in clinical samples from ATLL patients. Using RNA-seq data from CD4$^+$ T cells isolated from 66 ATLL patients and 3 healthy controls (Kataoka et al, 2015; Kogure et al, 2022), we found that both alpha-SMA (*ACTA2*) and ICAM-1 (*ICAM1*) were significantly upregulated in ATLL samples, whereas expression of gamma-actin (*ACTG1*), CD18 (*ITGB2*), and the housekeeping gene beta-glucuronidase (*GUSB*) remained unchanged (Fig. 8D). Together, these results demonstrate that hyperactive PLCG1 signaling in primary T cells mirrors findings in Hut78 cells and provides molecular insight into ATLL pathogenesis.

## Discussion

In this manuscript, we revealed that the mutated forms of PLCG1 found in ATLL enhanced pro-survival signaling of T lymphocytes. They enhanced LAT condensation to promote ERK activation. They also increased the expression of ICAM-1, which promotes ERK phosphorylation in neighboring cells through the integrin LFA1. The hyperactive ERK signaling rendered drug resistance to HDAC inhibitor vorinostat. We also discovered that PLCG1 mutants induced the expression of smooth muscle actin (alpha-SMA), which is normally restricted to smooth muscle cells. Alpha-SMA directly binds PLCG1 to enhance its phosphorylation and unleash the enzyme activity of PLCG1(Fig. 8E). We confirmed that PLCG1 mutations enhance the expression of alpha-SMA and ICAM-1 in human primary T cells and samples from ATLL patients. Together, our work extensively defined the cellular phenotypes of ATLL-associated PLCG1 mutations and provided an underlying molecular mechanism that explains their capacity to promote oncogenesis.

We exploited three common ATLL-related PLCG1 mutants to investigate hyperactive PLCG1 signaling. Among the three, the R48W mutation led to a mild increase in signaling, S345F showed an intermediate effect, and D1165H induced the strongest response.

These signaling patterns align with their known enzymatic activities and their ability to scaffold LAT condensation. Since these mutations are distributed across different regions of PLCG1, the consistent enhancement of pro-survival signaling, proliferation, and drug resistance observed in all mutants suggests that these phenotypes result from PLCG1 hyperactivation, rather than being specific to individual point mutations. As a future direction, it would be intriguing to determine if the different signaling strength, as triggered by R48W, S345F, or D1165H, correlates with disease progression and treatment outcomes.

Vorinostat is an FDA-approved drug for cutaneous T-cell lymphoma. However, the overall response rate is usually below 35% (Zinzani et al, 2016), and the mechanism is still incompletely understood. PLCG1 mutations were frequently discovered in cutaneous T-cel lymphoma(Chang et al, 2018; McGirt et al, 2015; Park et al, 2017; Vaque et al, 2014). Our findings that PLCG1 mutants rendered resistance to vorinostat suggested a mechanism explaining the resistance observed in clinics. Importantly, we showed that the ERK inhibitor can reverse the resistance to vorinostat, suggesting a strategy to improve the anti-tumor efficacy of vorinostat.

TCR-triggered native PLCG1 activation is transient (peaks around 1 min and then decays(Zeng et al, 2021)), which imposes a challenge in identifying components that promotes PLCG1 activation. The hyperactive mutants of PLCG1 enabled a constitutively activated PLCG1 form and provided a handle to identify the regulators of PLCG1 which might be missed in TCR-triggered transient activation. Through a pull-down assay, we identified smooth muscle actin as a positive regulator for handle activation.

We unexpectedly discovered that the expression of PLCG1 mutants induced the gene signature of smooth muscle contraction, which is typically restricted to smooth muscle cells. We showed that the signature gene alpha-SMA is specifically expressed in cells acquiring PLCG1 mutants. Alpha-SMA binds and activates PLCG1. This could form a positive loop to strengthen the hyperactive signaling in PLCG1 mutants.

In this study, we used two commonly used blood cancer cell lines Jurkat and Hut78 to investigate PLCG1 signaling. We also confirmed major results in human primary T cells. A whole-genome sequencing analysis revealed that the ATLL mutants were absent in the wild-type Jurkat and Hut78 cells (Fig. EV5). Meanwhile, Jurkat harbors a common variant T2438C (aa I813T) in PLCG1, which is present in 58.3% of healthy individuals. Hut78 harbors an additional common variant A835G (aa S279G), which is present in 15.4% of healthy individuals (Fig. EV5A). Based on the solved structure of PLCG1 (Hajicek et al, 2019), the residues at 813 and 279 are located far away from the catalytic core and the regulatory domain autoinhibitory interface, which is in contrast with the ATLL mutants. Neither of the two residues form chemical bonds with other residues. Using AlphaFold III

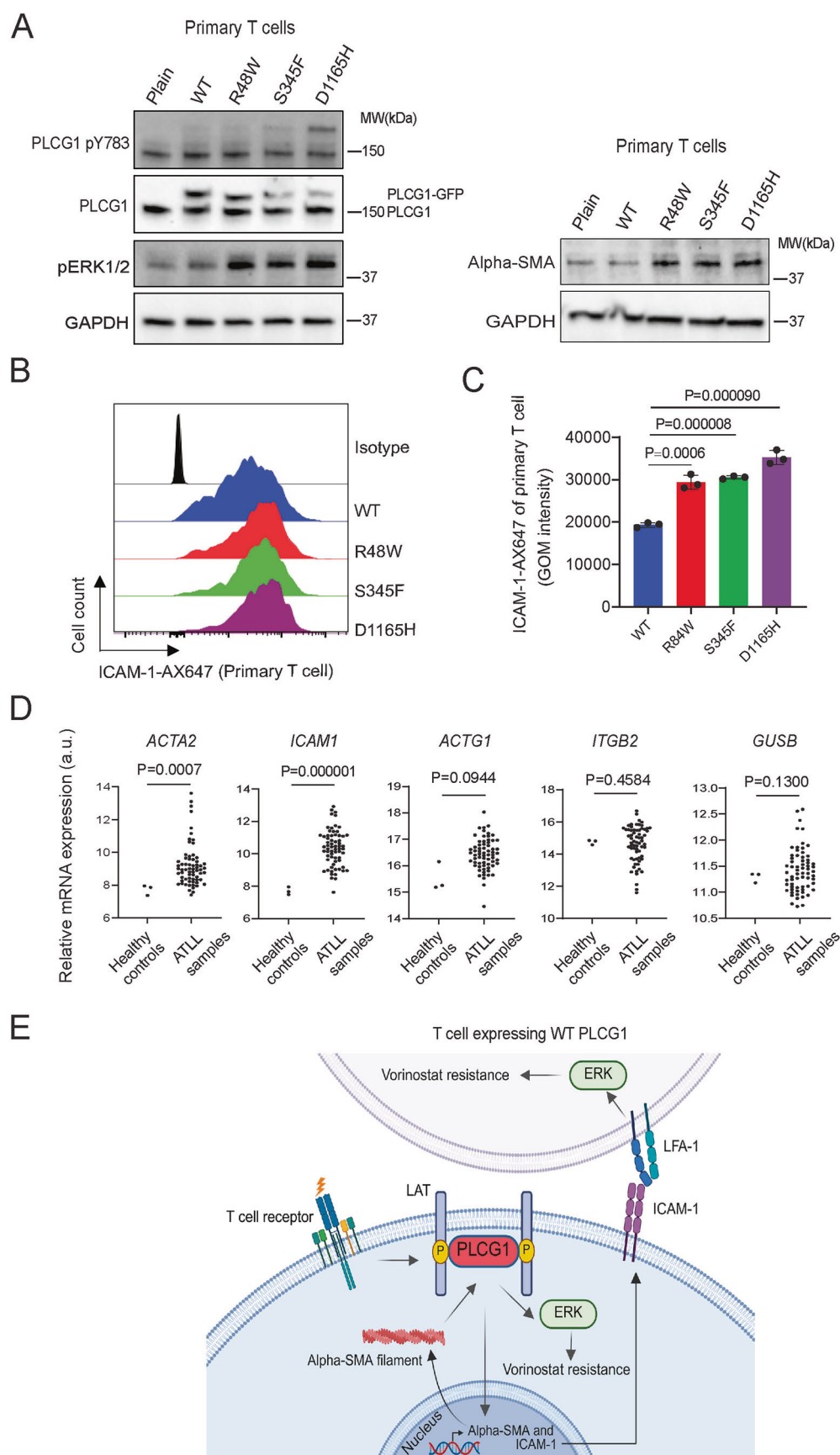

**Figure 8. Hyperactive PLCG1 signaling in human primary T cells and its clinical relevance to T-cell leukemia/lymphoma.**

(A) PLCG1 mutation induced ERK phosphorylation and alpha-SMA expression in human primary T cells as determined by Western blot. (B) PLCG1 mutation enhanced ICAM-1 expression in human primary T cells as determined by flow cytometry. (C) Quantification of ICAM-1 expression from (B). Shown are mean ± SD from $n = 3$ biological replicates. Unpaired two-tailed $t$ test was used. (D) RNA-seq analysis of ATLL samples. The expressions of alpha-SMA (*ACTA2*) and ICAM-1 (*ICAM1*) in ATLL samples ($n = 66$) were compared to those in healthy controls ($n = 3$). Unpaired two-tailed Welch's $t$ test was used. (E) Mechanisms and consequences of hyperactive PLCG1 signaling triggered by T-cell leukemia/lymphoma-associated mutations. Source data are available online for this figure.

prediction, I813T and S279G did not induce a significant change in local or global structure. These observations suggest that the two common variants are not likely to cause a significant change in the structure and activity of PLCG1.

# Methods

**Reagents and tools table**

| Reagent/resource | Reference or source | Identifier or catalog number |
|---|---|---|
| **Experimental models** | | |
| HEK293T | Vale lab at UCSF | CVCL_0063 |
| Jurkat T cell E6.1 | Vale lab at UCSF | CVCL_0367 |
| Hut78 | Müschen lab at Yale | CVCL_0337 |
| **Recombinant DNA** | | |
| pMD2.G | Su Lab at Yale | XSB395 |
| psPAX | Su Lab at Yale | XSB396 |
| pHR-PLCG1-sfGFP | This study | XSB434 |
| pHR-PLCG1 R48W-sfGFP | This study | XSB769 |
| pHR-PLCG1 S345F-sfGFP | This study | XSB770 |
| pHR-PLCG1 D1165H-sfGFP | This study | LZB135 |
| pET15b_His6-Thrombin-ybbR-alpha-SMA | This study | LZB176 |
| **Antibodies** | | |
| Anti-CD3 | eBioscience | CD3 Monoclonal Antibody (OKT3), Cat#16-0037-85 |
| Anti-CD28 | eBioscience | Cat#16-0289-85 |
| APC Anti-CD69 | Biolegend | Cat#310910 |
| Alex Fluor647 Anti-CD54 | Biolegend | Cat#353113 |
| APC Anti-CD18 | Biolegend | Cat#373405 |
| PE Anti-CD11a | Biolegend | Cat#350606 |
| APC Anti-CD49a | Biolegend | Cat#328313 |
| PE Anti-Integrin β5 | Biolegend | Cat#345203 |
| PE Anti-CD150 | Biolegend | Cat#306307 |
| Anti-GFP | Invitrogen | Cat#A11122, Pull down assay |
| Anti-CD18 | Biolegend | Cat#302102, Blocking antibody |
| Anti-pERK | Cell Signaling | Cat#9101, WB (1:2000) |
| Anti-PLCG1 | Cell Signaling | Cat#5690, WB (1:2000) |
| Anti-PLCG1 pY783 | Cell Signaling | Cat#2821, WB (1:2000) |
| Anti-alpha-SMA | R&D Systems | Cat#MAB1420, WB (1:5000) |

| Reagent/resource | Reference or source | Identifier or catalog number |
|---|---|---|
| Anti-beta-actin | Cell Signaling | Cat#3700, WB (1:5000) |
| Anti-gama-actin | Proteintech | Cat#11227-1-AP, WB (1:5000) |
| Anti-GAPDH | Biolegend | Cat#649202, WB (1:5000) |
| Anti-Mouse IgG | Invitrogen | Cat#31430, WB (1:15,000) |
| Anti-Rabbit IgG | Invitrogen | Cat#31460, WB (1:10,000) |
| **Chemicals, enzymes and other reagents** | | |
| Vorinostat | Cayman Chemical | Cat#10009929 |
| Belinostat | Cayman Chemical | Cat#34084 |
| Panobinostat | Cayman Chemical | Cat#13280 |
| LY3214996 | Cayman Chemical | Cat#27936 |
| Napabucasin | Cayman Chemical | Cat#22255 |
| Rhodamine-actin | Cytoskeleton | Cat#AR05-B |
| Human platelet non-muscle actin protein | Cytoskeleton | Cat#APHL99-C |
| BSA | Sigma | Cat#A9647-100G |
| KOD DNA Polymerase | EMD Millipore | Cat#71085-3 |
| Dynabeads | Thermo | Cat#11132D |
| Protein G Beads | Ocean NanoTech | Cat#MGP3000-002 |
| **Software** | | |
| GraphPad Prism 10.0.1 | https://www.graphpad.com/ | |
| ImageJ (Fiji) | https://imagej.net/ij | |
| FlowJo 10.10.0 | https://www.flowjo.com/ | |
| **Other** | | |
| CellTrace™ Far Red Cell Proliferation Kit | Invitrogen | Cat#C34572 |
| CellTrace™ Violet Cell Proliferation Kit | Invitrogen | Cat#C34571 |
| RNeasy Plus Mini Kit | Qiagen | Cat#74134 |
| IL-2 ELISA MAX™ Standard ELISA Kit | Biolegend | Cat#431801 |
| Human Latent TGF-β Pre-coated ELISA Kit | Biolegend | Cat#432907 |
| Human Phospho-Kinase Array Kit | Bio-techne | Cat#ARY003C |
| Cell Counting Kit-8 (CCK-8) | Dojindo | Cat#CK04 |

## Cell culture

HEK293T cells were cultured in DMEM medium (Gibco) supplemented with 10% fetal bovine serum (FBS, Gibco) and 1% Penicillin-Streptomycin-L-Glutamine (PSG, Corning). Jurkat and

Hut78 cell lines were maintained in RPMI 1640 medium (Gibco) with 10% FBS and 1% PSG. Cells were cultured at 37 °C in a humidified incubator with 5% $CO_2$.

## Recombinant protein purification

Human LAT (aa 1–233), Sos1 (aa 1117–1319), and Grb2 (aa 1–154) recombinant proteins were purified from BL21 (DE3) bacteria as described in our previous study (Su et al, 2016). Rat PLCG1 (aa 21–1215) WT, as well as PLCG1 R48W, S345F, and D1165H mutant proteins were expressed and purified as previous published protocol (Hajicek et al, 2019). The LAT protein remains an N-terminal His$_8$ tag, while Sos1, Grb2, and PLCG1 were removed N-terminal purification tags, such as GST and His$_6$.

The full-length human alpha-SMA gene (aa 1–377) gene was cloned into pET-15b vector and transformed into BL21 (DE3) competent cells to express His$_6$-ybbR-alpha-SMA. The cells were cultured in LB medium with Ampicillin to an $OD_{600}$ of 0.6–0.8 with shaking and then induced with 0.1 mM IPTG for 6 h at 37 °C. After induction, the cells were harvested by centrifugation and resuspended in 20 mM Tris-HCl, pH 8.0, 300 mM NaCl, 1 mM TCEP, 10% Glycerol containing protease inhibitor cocktail (Roche). Bacterial cells were lysed using a cell disruptor, and the lysate was treated with 0.5% Triton X-100, 10 μg/ml DNase, and 1 mM $MgCl_2$ with rotation at 4 °C for 1 h, followed by centrifugation at 25,000 rpm for 45 min at 4 °C. The alpha-SMA recombinant protein, associated with bacterial outer membranes, was pelleted and then solubilized with 0.5% N-Lauroylsarcosine sodium salt (Sigma) in a buffer containing 20 mM Tris-HCl (pH 8.0), 150 mM NaCl, 1 mM TCEP, and 10% glycerol. The soluble fraction was then purified using Ni-NTA beads, washed with buffer containing 20 mM imidazole (20 mM Tris-HCl, pH 8.0, 150 mM NaCl, 1 mM TCEP, 10% glycerol), and eluted with 250 mM imidazole in 20 mM Tris-HCl (pH 8.0), 150 mM NaCl, 1 mM TCEP, and 10% glycerol.

## Biochemical reconstitution and FRAP analysis of LAT condensate on supported lipid bilayers

PLCG1-driven LAT condensation on lipid bilayers was performed as described previously (Zeng et al, 2021). Briefly, 96-well glass-bottom imaging plate was cleaned with 5% Hellmanex III (Sigma) overnight, then washed three times with 5 M NaOH at 50 °C. The plate was thoroughly rinsed with ddH2O and PBS. Supported NTA lipid bilayers, for anchoring with his-tagged recombinant protein, were formed by adding 20 μl of small unilamellar vesicles (SUVs) containing 2% DOGS-NTA, 0.1% PEG-5000, and 99.8% POPC lipids (Avanti) to 200 μl PBS in each well, incubating for 1 h at 37 °C. Excess SUVs were removed by washing with basic buffer (50 mM HEPES, pH 7.4, 150 mM NaCl, 1 mM TCEP). Lipid bilayers were blocked with freshly prepared clustering buffer (50 mM HEPES, pH 7.4, 150 mM NaCl, 1 mM TCEP, 1 mg/ml BSA) for 30 min at 37 °C. His8-tagged pLAT-C3B was then incubated with the bilayers for 3 h, followed by washing with clustering buffer to remove unanchored pLAT. LAT condensation on supported lipid bilayers was triggered by adding PLCG1, Grb2, and Sos1 recombinant proteins, diluted in an oxygen scavenger solution (0.2 mg/ml glucose oxidase, 0.035 mg/ml catalase, 70 mM beta-mercaptoethanol, and 25 mM glucose) in clustering buffer, and incubated for 30 min at 37 °C. LAT clusters were imaged using

a Nikon Ti2-E inverted fluorescence motorized microscope with a 100x TIRF objective. Cluster was photobleached, and fluorescence recovery was monitored over 5 min at 2-s intervals using TIRF microscopy. Clustering and FRAP data were analyzed with ImageJ. The LAT clustering level was quantified using normalized variance ($SD^2$/mean), where the mean represents the average fluorescence intensity after background subtraction. FRAP half-recovery time and recovery percentage were calculated using GraphPad Prism 10.0.1 software, fitted to a one-phase association model.

## Live cell imaging of LAT condensation in Jurkat T cells

The 96-well glass-bottom imaging plate was coated with 100 μl of 5 μg/ml anti-CD3 monoclonal antibody OKT3 (eBioscience) in PBS and incubated overnight at room temperature. Unbound OKT3 antibody was then removed by washing with PBS. The plate was equilibrated in 100 μl of imaging medium, consisting of phenol red-free RPMI medium (Gibco) with 20 mM HEPES, pH 7.4. Jurkat T cells expressing LAT-mCherry and PLCG1-GFP were harvested and resuspended in imaging medium. TIRF microscopy was used to capture real-time LAT clustering upon adding 100 μl of 1 million/ml Jurkat cells to the OKT3-coated well at 37 °C. Data analysis was performed using ImageJ.

## Human primary T cell growth assay

Human peripheral blood mononuclear cells (PBMC) were purchased from Zen-Bio company. Primary T cells were isolated using the EasySepTM human T cell isolation kit (Stemcell technologies) according to the manufacturer's instructions. Primary T cells were co-cultured with Human T-activator CD3/CD28 Dynabeads at a 1:1 ratio in RPMI medium (Gibco) supplemented with 10% FBS, 1% PSG, 10 mM HEPES, pH 7.4 (Gibco), 2 mM L-Glutamine (Gibco), 1× MEM-NEAA (Gibco), 0.55 mM 2-mercaptoethanol (Gibco), and 200 U/ml IL-2 (PeproTech) for 3 days. PLCG1-GFP (WT or mutant) lentivirus vectors (3000 ng) was co-transfected with packaging plasmids pMD2.G (500 ng) and psPAX (500 ng) into 293 T cells using 12 μl PEI (Polysciences) in a 6-well plate for 48 h. The lentivirus was then collected and used to infect the primary T cells/Dynabeads mixture in the presence of 5 μg/ml polybrene (Santa cruz biotechnology) by plate centrifugation (800 × g, 90 min at 32 °C). After 4 days of lentiviral infection, Dynabeads were removed using a magnetic separator (BioLegend), and the infected primary T cells were rested for 5 days to test the T cell growth.

For the human primary T cell growth assay, 0.25 million/ml T cells were seeded into a 96-well U-bottom plate (Corning) in 200 μl primary T cell culture medium. Cell counts were performed using an automated cell counter (Bio-Rad).

## Flow cytometry

Cell surface marker detection in T cells: T cells were collected and washed with PBS and sorting buffer (0.5% BSA, 2 mM EDTA in PBS). The cells were stained with fluorescence conjugated antibody on ice for 30 min, then washed with sorting buffer. Stained cells were analyzed using a BD FACS machine, and FlowJo software was used for quantifying the FACS data.

Calcium Flux Monitoring in Jurkat Cells: Jurkat T cells stably expressing the calcium sensor GCaMP6s and either wild-type or

mutant PLCG1-GFP were collected and resuspended in RPMI medium without phenol red (Gibco), supplemented with 20 mM HEPES, pH 7.4. The cells were loaded into the FACS machine, where their pre-activation status was recorded at low speed for 1 min. Subsequently, the cells were activated with 10 µg/ml OKT3 antibody and recorded for an additional 5 min at low speed. Calcium signals were captured in the DsRed channel and normalized to the average intensity of the pre-activation status for each group.

Apoptosis Analysis in Hut78 Cells: Vorinostat-treated Hut78 cells were collected and washed with PBS and Annexin V binding buffer (10 mM HEPES, pH 7.4, 140 mM NaCl, 2.5 mM CaCl₂). Annexin V (BioLegend) was diluted 1:100 in the binding buffer and used to stain the cells for 15 min at room temperature. The cells were then analyzed directly by FACS without further washing.

## Western blot and phospho-kinase profiling

Jurkat cells were washed with PBS and rested for 30 min at 37 °C. The cells were stimulated with 2 µg/ml OKT3 and 2 µg/ml Anti-CD28 antibody (eBioscience) for 2 min at 37 °C. SDS-PAGE loading buffer (Bio-Rad) containing protease inhibitors cocktail (Roche) was added to stop the activation and lyse the cells. The cell lysate was boiled for 10 min at 100 °C and then centrifuged at 1,4000 rpm for 5 min. The protein supernatant was loaded into 4–20% gradient gel (Bio-Rad) for SDS-PAGE and subsequently transferred to a PVDF membrane (Bio-Rad). The membrane was blocked with 5% nonfat milk in TBST buffer for 1 h at room temperature. It was then incubated overnight at 4 °C with primary antibodies against PLCG1 (CST), PLCG1 pY783 (CST), pERK1/2 pT202/pY204 (CST), and GAPDH (Biolegend) diluted in 3% BSA TBST buffer. The membrane was washed three times with TBST buffer to remove unbound primary antibodies. The membrane was incubated with horseradish peroxidase (HRP)-conjugated secondary antibody (Thermo Scientific) for 1 h at room temperature. TBST buffer was used to wash out unbound secondary antibody for three times. Enhanced chemiluminescent (ECL) HRP substrate (Thermo Scientific) was applied to the membrane for 3 min and then imaged using a chemiDoc imaging system (Bio-Rad). Image lab (Bio-Rad) was utilized to quantify protein expression.

Hut78 cells stably expressing PLCG1-GFP WT, or the mutant were washed with PBS and lysed using SDS-PAGE loading buffer containing a protease inhibitor cocktail. The remaining steps were conducted as described above. The Proteome Profiler Human Phospho-Kinase Array Kit (R&D Systems) was used to analyze the phosphorylation of 37 kinases in Hut78 cells, following the manufacturer's instructions.

## Enzyme-linked immunosorbent assay (ELISA)

Hut78 cells expressing PLCG1-GFP WT, or the mutant were seeded in 24-well plate at a density of 0.4 million/ml in 500 µl of RPMI cell culture medium and incuabted for 72 h. The cells were spun down, and the supernatant was used for ELISA analysis of IL-2 and latent TGF-beta secretion levels by the kits from Biolegend according to manufacturer's instructions.

## Cell aggregation assay

Hut78 cells were seeded in 96-well flat-bottom plate with 0.1 million cells per well in 200 µl of RPMI cell culture medium and incubated for 48 h. The EVOS cell imaging system (Thermo) was used to record cell aggregates at ×4 objective.

## Cell viability analysis

Hut78 cells were seeded in 96-well plate at a cell density of 0.4 million/ml in 100 µl of RPMI cell culture medium and incubated overnight. Drugs were diluted into 100 µl of RPMI cell culture medium and used to treat the cells for 72 h. Cell counting kit-8 (CCK8, Dojindo) was then added to each well at a volume of 10 µl, followed by a 2 h incubation at 37 °C. The optical density at 450 nm (OD450) was recorded by a plate reader (Molecular Devices).

## Pull-down assay

Hut78 cells expressing PLCG1-GFP WT, or the mutant were washed for three times with PBS and then lysed in 1% Triton X-100 in 20 mM Tris, pH 8.3, 150 mM NaCl buffer containing protease inhibitor cocktail, with vertexing for 30 min at 4 °C. The lysate was centrifuged at 15,000 rpm for 15 min at 4 °C, and the supernatant was collected and quantified for total protein concentration using Bradford assay with Bio-Rad protein assay dye reagent concentrate. MonoMag protein G beads (Ocean nanotech) were washed three times with 0.25% Triton X-100 in 20 mM Tris-HCl, pH 8.3, 150 mM NaCl. The protein G beads, anti-GFP antibody (Invitrogen), and 500 µg of cell lysate were co-incubated with rotation overnight at 4 °C. The beads were further washed three times with 0.25% Triton X-100 in 20 mM Tris-HCl, pH 8.3, 150 mM NaCl. The pull-down binding proteins on beads were mixed with loading buffer (Bio-Rad). The beads mixture was loaded into 4-20% gradient gel for SDS-PAGE analysis. Blue safe protein stain (Thermo) reagent was used to stain the gel. The specific band was cut and stored in 1% acetic acid for Mass Spectrometry analysis at the Protein Facility of the Iowa State University Office of Biotechnology.

## Actin binding partner spin-down assay

Recombinant alpha-SMA protein was buffer-exchanged to 5 mM Tris-HCl, pH 8.0, 0.2 mM CaCl2, 1 mM TCEP, 1 mM DTT using Cytiva PD SpinTrap G-25 column (GE). Alpha-SMA protein was polymerized into filaments (F-actin) together with Rhodamine muscle actin (Cytoskeleton) by adding 10× polymerization buffer (250 mM Tris-HCl, pH 7.2, 500 mM KCl, pH 8.0, 20 mM MgCl₂) and incubating for 1 h at room temperature. The F-actin pellet was spun down at 83,000 × $g$ for 30 min in a high-speed centrifuge and resuspended in 50 mM HEPES, pH 7.4, 150 mM NaCl, 1 mM TCEP. F-actin, PLCG1, and Myosin proteins were co-incubated in 50 µl of 50 mM HEPES, pH 7.4, 150 mM NaCl, 1 mM TCEP for 2 h at room temperature. A high-speed centrifuge was then used to pellet the F-actin binding partners. The supernatant and pellet were loaded into 4–20% gradient gel for SDS-PAGE analysis. Blue safe protein stain (Thermo) reagent was used to stain the gel.

## Quantitative real-time PCR assay

Quantitative real-time PCR (qPCR) was performed to measure ICAM-1 mRNA expression. Hut78 cells were harvested, and total RNA was extracted using RNeasy Plus Mini Kit (Qiagen). The cDNA was further generated with RevertAid First Strand cDNA Synthesis Kit (Thermo)

according to the manufacturer's protocol. The qPCR reaction was carried out using PowerTrack™ SYBR Green Master Mix (Applied Biosystems). The ICAM-1 specific primers used were: forward 5'-AGCGGCTGACGTGTGCAGTAAT-3' and reverse 5'-TCTGA-GACCTCTGGCTTCGTCA-3'. GAPDH was used as the internal reference gene. Relative expression levels of ICAM-1 were calculated using the $2^{-\Delta\Delta Cq}$ method.

## Gene expression analysis of ATLL patients

We re-analyzed our previously published RNA-seq data from 69 samples, comprising 66 ATLL samples and 3 peripheral blood CD4 + T-cell samples from healthy controls (Kataoka et al, 2015; Kogure et al, 2022). Mapping was performed using Genomon pipeline v2.6.2 (https://github.com/Genomon-Project/), with a custom reference genome based on 1000genomes Reference Genome Sequence (hs37d5) supplemented with HTLV-1 genome (AB513134). Raw read counts for Gencode v29lift37 genes were obtained using the featureCounts command in Subread v2.0.0 (Liao et al, 2014), and normalization was performed using the median of ratios method implemented in DESeq2 v.1.44.0 (Love et al, 2014). Genes with more than five read counts in at least 20% of the samples (a total of 28,666 genes) were retained after filtering. Subsequently, variance-stabilizing transformation (VST) was applied to the normalized counts. VST-transformed expression values were compared using unpaired two-tailed Welch's $t$ test.

## Statistical analysis

Unpaired two-tailed $t$ test and Welch's $t$ test were used to compare differences between two groups using GraphPad Prism version 10.0.1.

# Data availability

RNA-Seq data: NCBI Gene Expression Omnibus GSE280396. Whole-genome sequencing of Jurkat cells (Accession: SRX 28560196) and Hut78 cells (Accession: SRX28560197): NCBI SRA database PRJNA1255536.

The source data of this paper are collected in the following database record: biostudies:S-SCDT-10_1038-S44319-025-00546-x.

# Peer review information

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

## Acknowledgements

We thank Markus Müschen at Yale University for providing Hut78 cells. We thank Diane Krause at Yale University for discussion of the work. Supercomputing resources were provided by the Human Genome Center, Institute of Medical Science, The University of Tokyo, Tokyo, Japan. XS was supported by an American Cancer Society Research Scholar Grant 135926, the Rally Foundation A Collaborative Pediatric Cancer Research Awards Program 22YIC53, the Yale SPORE in skin cancer DRP Award P50 CA121974, the Yale Cancer Center Pilot Award, the Yale DeLuca Pilot Award, the NIGMS MIRA program R35 GM138299, the Gabrielle's Angel Foundation Medical Research Award, the Pershing Square Sohn Prize for Young Investigators in Cancer research, the NCI Exploratory/Developmental Research Grant R21 CA286364, the NIH Exploratory/Developmental Bioengineering Research Grants (EBRG) R21 CA294038, the Human Frontier Science Program Early-Career Research Grant RGY0088/2021, the Yale Liver Center Pilot Award P30 DK034989, the Yale Lion Heart Pilot Grant, and the NIH Director's Transformative Research Award EB037112. LZ was supported by the CRI-Irvington Postdoctoral Fellowship (CRI3516). XZ was supported by the Leslie Warner Postdoctoral Fellowship. KS was supported by the Daiichi Sankyo Foundation of Life Science Fellowship. NH and JS acknowledge support from the NIH (R35 GM149299, R01 CA258993, P30 CA016086, and R56 AG083424). KK was supported by Japan Society for the Promotion of Science (JSPS) KAKENHI (JP21H05051 (KK)).

## Author contributions

**Longhui Zeng**: Conceptualization; Data curation; Formal analysis; Funding acquisition; Validation; Investigation; Visualization; Methodology; Writing—original draft; Project administration; Writing—review and editing. **Xinyan Zhang**: Data curation; Formal analysis; Funding acquisition; Validation; Methodology. **Yiwei Xiong**: Data curation; Formal analysis; Methodology. **Kazuki Sato**: Data curation; Formal analysis; Funding acquisition; Methodology. **Nicole Hajicek**: Data curation; Formal analysis; Methodology. **Yasunori Kogure**: Data curation; Formal analysis; Validation; Investigation; Visualization; Methodology; Writing—original draft. **Keisuke Kataoka**: Conceptualization; Data curation; Supervision; Funding acquisition; Validation; Methodology; Project administration; Writing—review and editing. **Seishi Ogawa**: Resources; Supervision; Writing—review and editing. **John Sondek**: Resources; Supervision; Funding acquisition; Writing—review and editing. **Xiaolei Su**: Conceptualization; Resources; Formal analysis; Supervision; Funding acquisition; Validation; Investigation; Visualization; Methodology; Writing—original draft; Project administration; Writing—review and editing.

Source data underlying figure panels in this paper may have individual authorship assigned. Where available, figure panel/source data authorship is listed in the following database record: biostudies:S-SCDT-10_1038-S44319-025-00546-x.

## Disclosure and competing interests statement

The authors declare no competing interests.

# Expanded View Figures

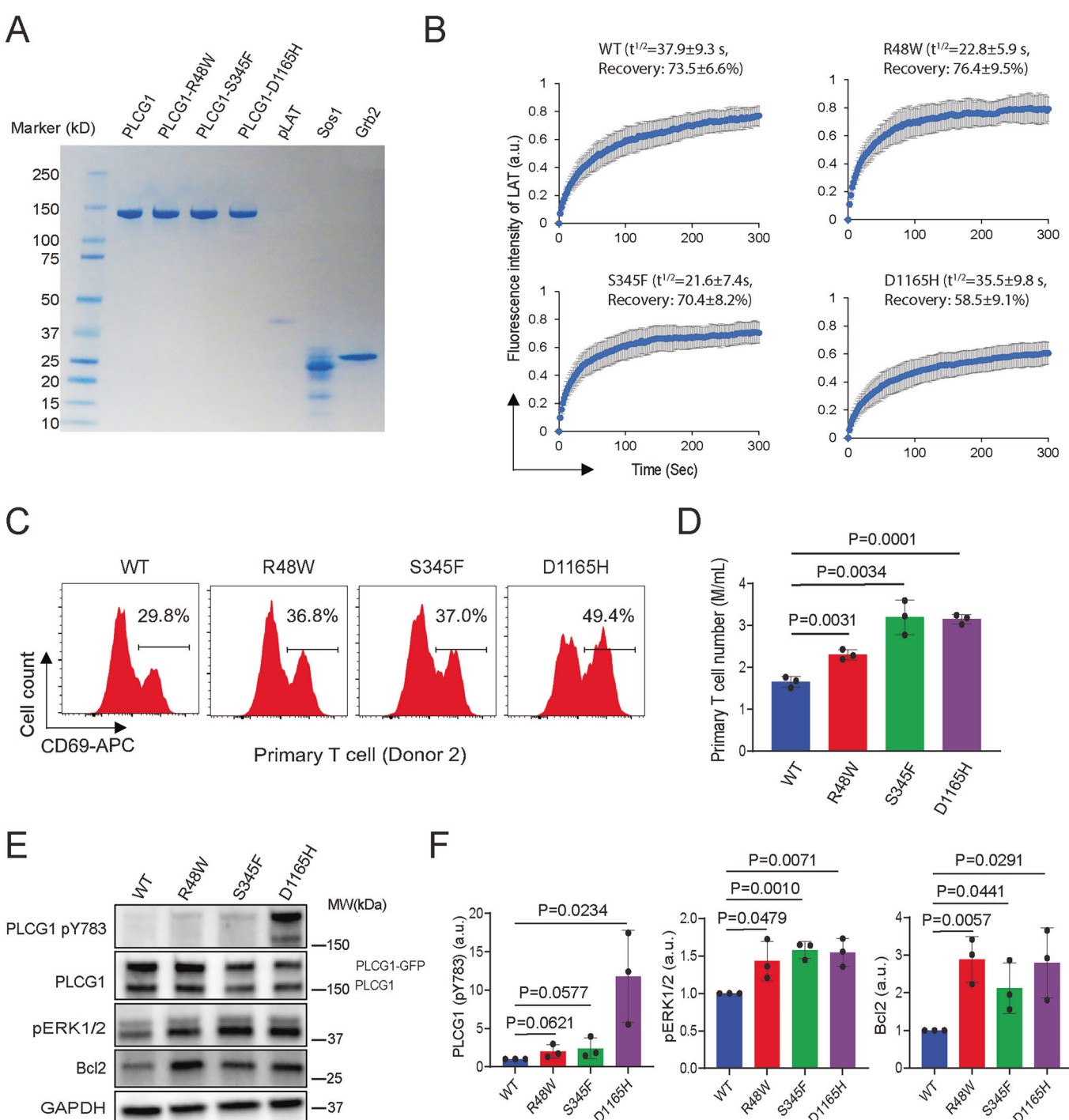

**Figure EV1. PLCG1 mutants promote the activation of T cells.**

(A) Recombinant proteins used in this study. The purified proteins were loaded to SDS-PAGE followed by Coomassie Blue staining. (B) FRAP analysis of LAT condensates. Shown are mean ± SD from $n = 10$ condensates. (C) Activation of human primary T cells expressing PLCG1 WT or mutants. The expression of CD69 was determined by flow cytometry 14 days after T cells were infected with lentivirus encoding PLCG1 WT or mutants. This is a repeated experiment using T cells from a different donor than what was used in Fig. 2G. (D) Proliferation of human primary T cells expressing PLCG1 WT or mutants. The cell number was quantified 14 days after T cells were infected with lentivirus encoding PLCG1 WT or mutants. This is a repeated experiment using T cells from a different donor than what was used in Fig. 2H. Shown are mean ± SD from $n = 3$ biological replicates. Unpaired two-tailed $t$ test was used. (E) Immunoblot analysis of signaling in Hut78 cells ectopically expressing GFP-tagged PLCG1 WT or mutants (without TCR activation). Low-titer virus was used so that PLCG1 WT or mutants were expressed at a similar level to the endogenous PLCG1. (F) Quantification of (E). Shown are mean ± SD from $n = 3$ biological replicates. Unpaired two-tailed t-test was used.

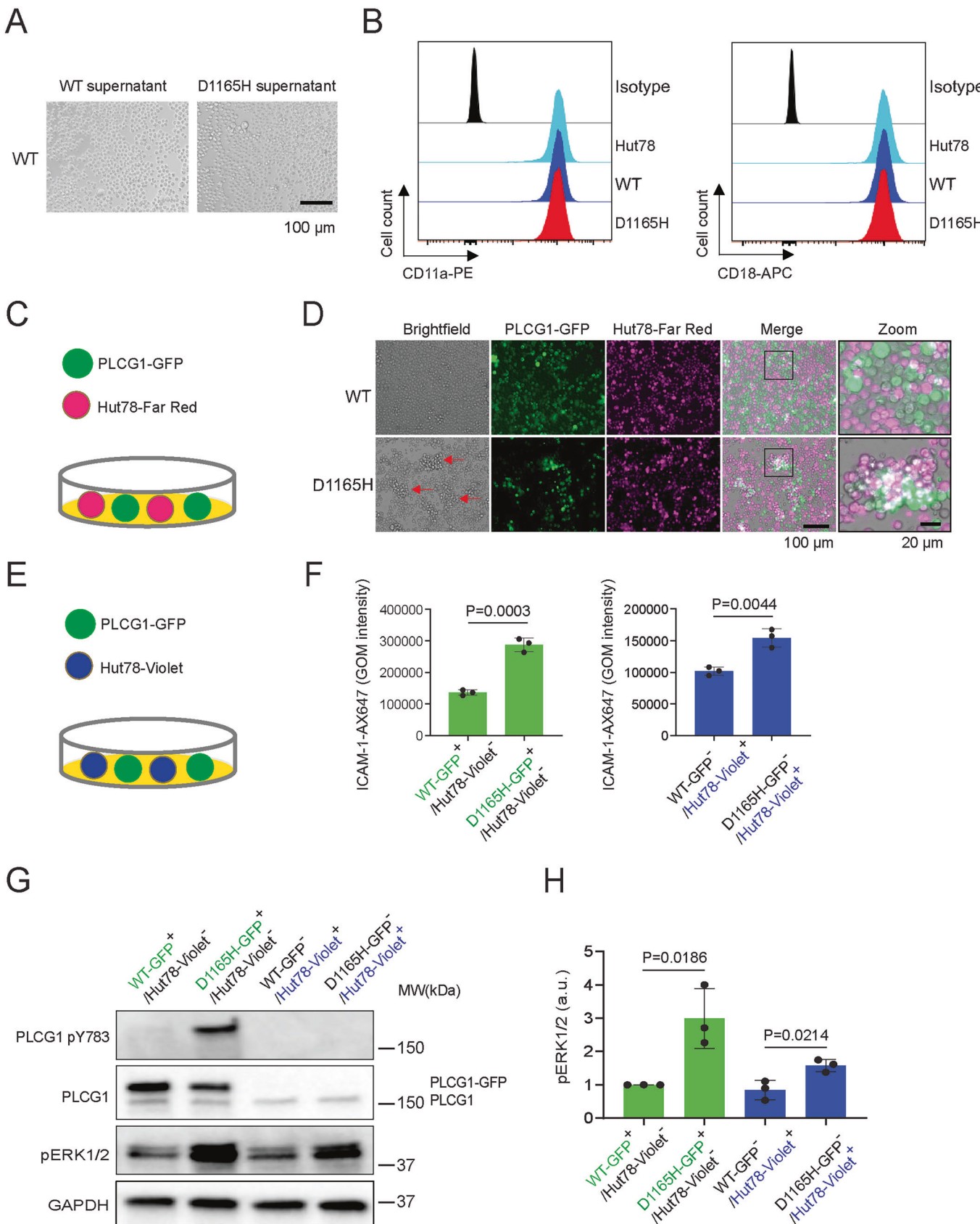

**Figure EV2. Hut78 cells expressing PLCG1 mutations induce aggregation and activation of neighboring cells expressing the wild-type PLCG1.**

(A) Conditioned media from Hut78 cells expressing D1165H did not induced cell aggregation. Hut78 cells expressing PLCG1 WT were cultured in supernatant from Hut78 cells expressing the WT or D1165H PLCG1 for 2 days. (B) The expression of LFA-1 (Two subunits, CD11a and CD18) on the cell surface of Hut78 was detected by flow cytometry. (C) Schematics of co-culture assay. Hut78 cells harboring PLCG1 WT or D1165H were co-cultured with Far-red dye-labeled plain Hut78 cells at 1:1 ratio. (D) Plain Hut78 cells co-aggregated with Hut78 expressing D1165H. Red arrow indicates larger cell aggregate. Scale bar: 100 μm. An enlarged inset is shown on the right. Scale bar: 20 μm. (E) Schematics of the co-culture assay. Violet labeled Hut78 cells were co-cultured with Hut78 cells expressing PLCG1-GFP WT or D1165H in a 1:1 ratio for 72 h. (F) ICAM-1 cell surface expression by FACS. Shown are mean ± SD from $n = 3$ biological replicates. Unpaired two-tailed t-test was used. (G) The co-culture cells were sorted by FACS, lysed, and analyzed by Western blot. (H) Quantification of ERK phosphorylation from (G). Shown are mean ± SD from $n = 3$ biological replicates. Unpaired two-tailed t-test was used.

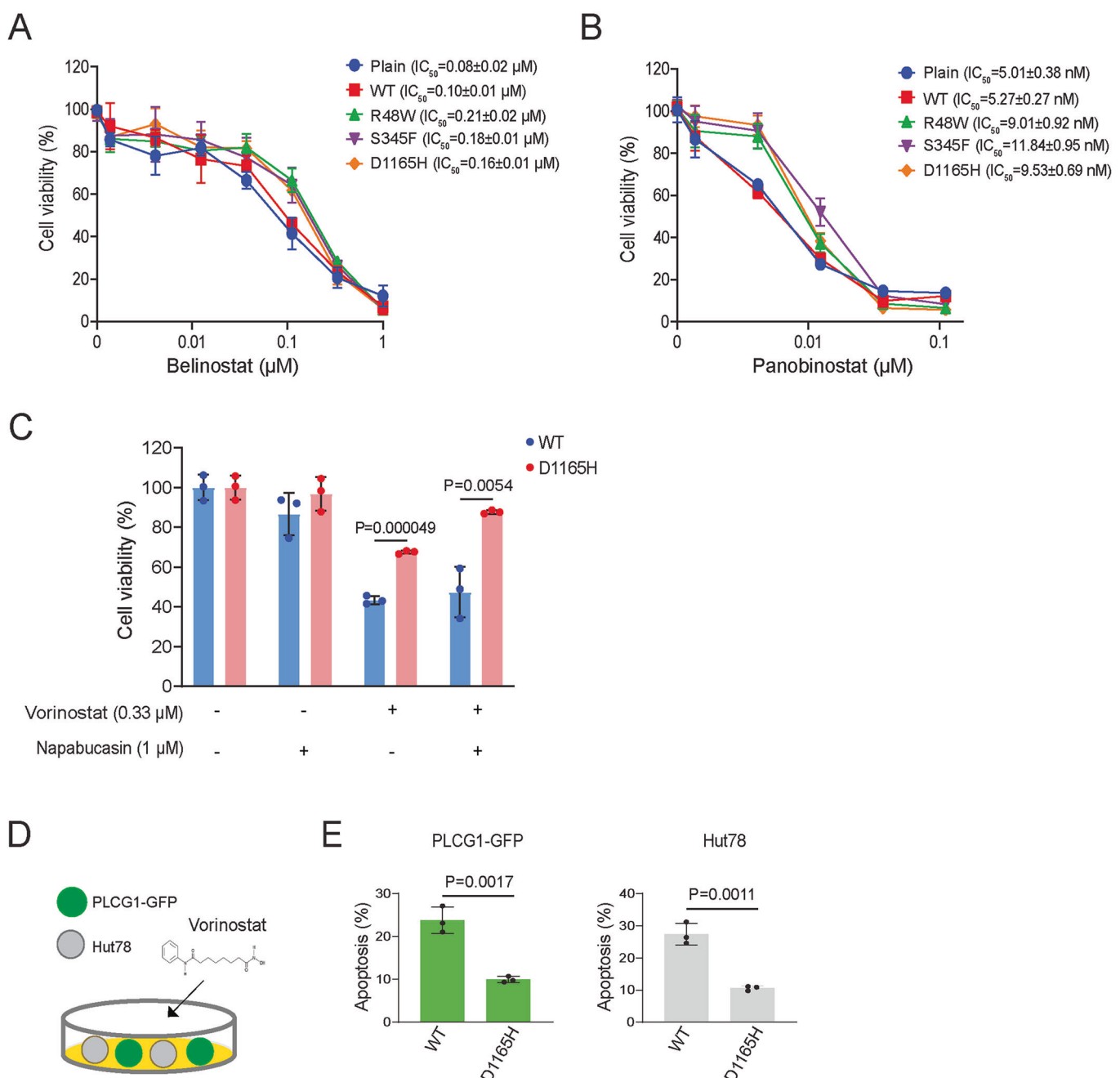

**Figure EV3. PLCG1 mutations confer Hut78 resistance to HDAC inhibitors.**

(A) PLCG1 mutations conferred Hut78 resistance to belinostat. The plain group is Hut78 cells without ectopically expressed PLCG1. The CCK8 assay was used to detect viable cell number after belinostat treatment for 72 h. Shown are mean ± SD from $n = 3$ biological replicates. (B) PLCG1 mutations conferred Hut78 resistance to panobinostat. The plain group is Hut78 cells without ectopically expressed PLCG1. The CCK8 assay was used to detect viable cell number after panobinostat treatment for 72 h. Shown are mean ± SD from $n = 3$ biological replicates. (C) STAT3 inhibitor napabucasin did not affect resistance to vorinostat. Shown are mean ± SD from $n = 3$ biological replicates. Unpaired two-tailed t-test was used. (D) Schematics of the co-culture assay with vorinostat treatment. Plain Hut78 cells (grey) were co-cultured with Hut78 cells expressing PLCG1-GFP WT or D1165H in a 1:1 ratio for 1 day, and then treated with 1 μM vorinostat for 48 h before being analyzed for apoptosis marker. (E) Hut78 cells expressing PLCG1 D1165H protected the neighboring plain Hut78 cells from vorinostat-induced apoptosis. Hut78 cells expressing PLCG1 WT or D1165H were co-cultured with plain Hut78 cells at 1:1 ratio. The apoptosis level, as indicated by annexin V staining, was determined by flow cytometry. Shown are mean ± SD from $n = 3$ biological replicates. Unpaired two-tailed t-test was used.

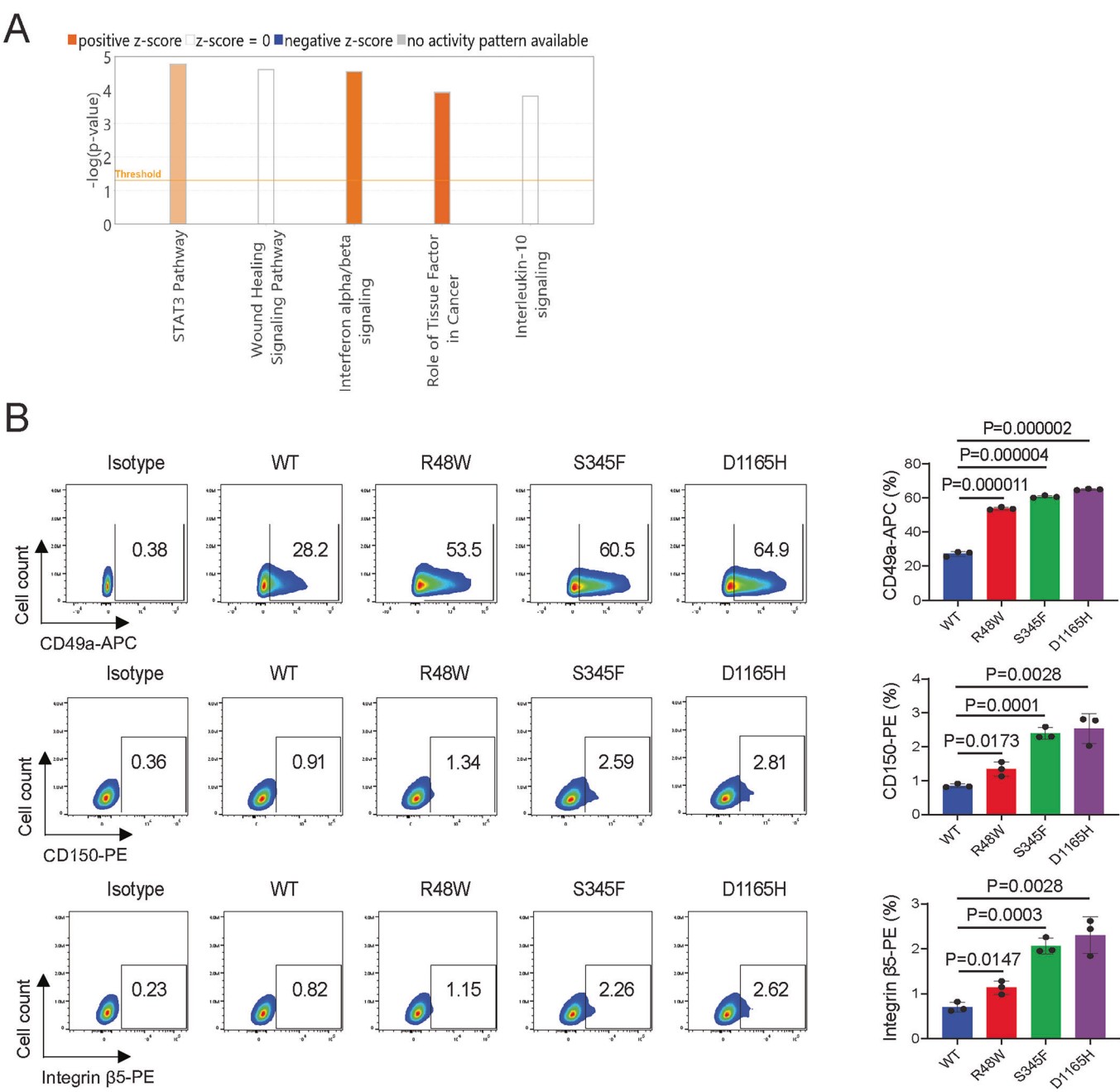

**Figure EV4. Overlapped genes between hyperactive PLCG1 signaling and TCR signaling.**

(A) Qiagen ingenuity pathway analysis (IPA) showed pathways enriched in overlapping genes between hyperactive PLCG1 signaling and TCR signaling. (B) Cell surface protein expression by flow cytometry. Shown are mean ± SD from $n = 3$ biological replicates. Unpaired two-tailed $t$ test was used.

A

| Cell line | Gene variant | Codon Change | Mutation type | Amino acid mutation | Allele frequency-gnomAD database | dbSNP ID |
|-----------|--------------|--------------|---------------|---------------------|-----------------------------------|----------|
| Jurkat | PLCG1-T2438C | aTc/aCc | T(0%)/C(100%), Homozygous | PLCG1-I813T | 58.3%, Common variant (>5%) | rs753381 |
| Hut78 | PLCG1-A835G | Agc/Ggc | A(43%)/G(57%), Heterozygous | PLCG1-S279G | 15.4%, Common variant (>5%) | rs2228246 |
| Hut78 | PLCG1-T2438C | aTc/aCc | T(43%)/C(57%), Heterozygous | PLCG1-I813T | 58.3%, Common variant (>5%) | rs753381 |

B

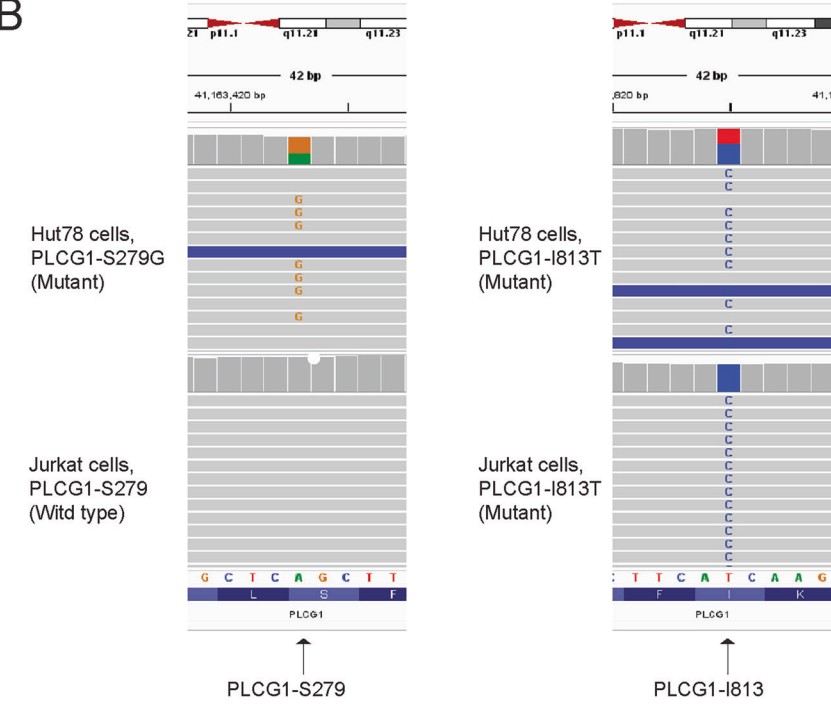

C                    D                    E

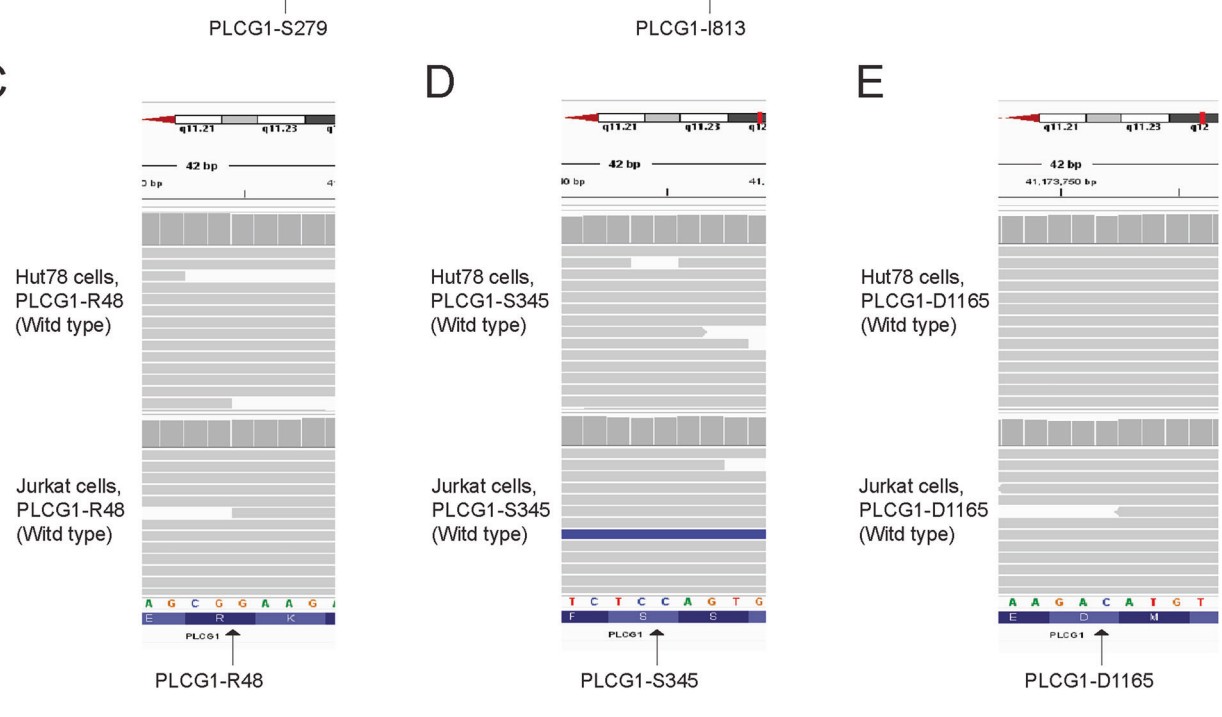

◀ **Figure EV5. PLCG1 sequence in Jurkat and Hut78 cells as revealed by whole-genome sequencing.**

(A) Common variants of PLCG1 in Jurkat and Hut78 cells. (B) Local view of the PLCG1 common variants in Jurkat and Hut78 cells as compared to reference human genome sequence. (C) Local view of sequences encoding PLCG1-R48. (D) Local view of sequences encoding PLCG1-S345. (E) Local view of sequences encoding PLCG1-D1165.

