## [Peer Review File · EMBO Reports]

Hyperactive PLCG1 induces cell-autonomous and bystander T cell activation and drug resistance

Longhui Zeng, Xinyan Zhang, Yiwei Xiong, Kazuki Sato, Nicole Hajicek, Yasunori Kogure, Keisuke Kataoka, Seishi Ogawa, John Sondek, and Xiaolei Su

Corresponding author(s): Xiaolei Su (xiaolei.su@yale.edu)

Review Timeline:

Transfer Date:	29th Jan 25
Editorial Decision:	31st Jan 25
Revision Received:	22nd May 25
Editorial Decision:	8th Jul 25
Revision Received:	14th Jul 25
Accepted:	28th Jul 25

Editor: Achim Breiling

Transaction Report: This manuscript was transferred to EMBO reports following peer review at The EMBO Journal.

Dear Dr. Su,

Thank you for transferring your manuscript to EMBO reports. I now went through the manuscript and the referee reports from The EMBO Journal (attached again). The referees have several comments, concerns, and suggestions to improve the manuscript, indicating that a major revision of the manuscript is necessary to allow publication of the study.

I thus invite you to revise your manuscript accordingly with the understanding that all concerns must be addressed in the revised manuscript and/or in a detailed point-by-point response.

EMBO Reports emphasizes novel functional insight with clear in vivo relevance over detailed mechanistic insight. Thus, EMBO Reports will not require addressing points regarding more mechanism experimentally, but welcomes such data, in case you already have it. However, it will be necessary that during the revision you address all points questioning the main conclusions of the study, and all technical concerns, or points regarding the experimental designs, model systems used, or data presentation.

In this case, EMBO reports does not need data from freshly isolated adult T-cell leukemia/lymphoma samples (referee #1), and - as already mentioned - does not require further mechanistic insight. However, other referee points that refer to the current conclusions of the study need attention, in particular the second point of referee #2 in order to show that the endogenous copies of Plcg1 in the used cell lines are indeed wild type.

Acceptance of your manuscript will depend on a positive outcome of another round of review at EMBO reports, using the same referees.

- 1) a .docx formatted version of the final manuscript text (including legends for main figures, EV figures and tables), but without the figures included. Please make sure that changes are highlighted to be clearly visible. Figure legends should be compiled at the end of the manuscript text.
- 2) individual production quality figure files as .eps, .tif, .jpg (one file per figure), of main figures and EV figures. Please upload these as separate, individual files upon re-submission. Please make sure that all figure panels are called out separately and sequentially in the manuscript text

For more details please refer to our guide to authors:
<http://www.embopress.org/page/journal/14693178/authorguide#manuscriptpreparation>

See also our guide for figure preparation:
http://wol-prod-cdn.literatumonline.com/pb-assets/embo-site/EMBOPress_Figure_Guidelines_061115-1561436025777.pdf

Moreover, please consult our guidelines for figure legend preparation:
<https://www.embopress.org/page/journal/14693178/authorguide#figureformat>

4) a complete author checklist, which you can download from our author guidelines (<https://www.embopress.org/page/journal/14693178/authorguide>). Please insert page numbers in the checklist to indicate where the requested information can be found in the manuscript. The completed author checklist will also be part of the RPF.

Please also follow our guidelines for the use of living organisms, and the respective reporting guidelines: <http://www.embopress.org/page/journal/14693178/authorguide#livingorganisms>

5) that primary datasets produced in this study (e.g. RNA-seq, ChIP-seq and array data) are deposited in an appropriate public database. This is now mandatory (like the COI statement). If no primary datasets have been deposited in any database, please state this in this section (e.g. 'No primary datasets have been generated and deposited').

The accession numbers and database should be listed in a formal "Data Availability " section (placed after Materials & Methods) that follows the model below. Please note that the Data Availability Section is restricted to new primary data that are part of this study.

Data availability

8) Regarding data quantification and statistics, please make sure that the number "n" for how many independent experiments were performed, their nature (biological versus technical replicates), the bars and error bars (e.g. SEM, SD) and the test used to calculate p-values is indicated in the respective figure legends (also for potential EV figures and all those in the final Appendix). Please also check that all the p-values are explained in the legend, and that these fit to those shown in the figure. Please provide statistical testing where applicable. Please avoid the phrase 'independent experiment', but clearly state if these were biological or technical replicates. Please also indicate (e.g. with n.s.) if testing was performed, but the differences are not significant. In case n=2, please show the data as separate datapoints without error bars and statistics.

See also:

<http://www.embopress.org/page/journal/14693178/authorguide#statisticalanalysis>

If n<5, please show single datapoints for diagrams. Please add to each legend (main, EV figures, Appendix, where applicable) a 'Data Information' section explaining the statistics used or providing information regarding replicates and scales. See: <https://www.embopress.org/page/journal/14693178/authorguide#figureformat>

9) Please add scale bars of similar style and thickness to any microscopic images, using clearly visible black or white bars (depending on the background). Please place these in the lower right corner of the images themselves. Please do not write on or near the bars in the image but define the size in the respective figure legend.

10) Please note our reference format:

11) We updated our journal's competing interests policy in January 2022 and request authors to consider both actual and perceived competing interests. Please review the policy <https://www.embopress.org/competing-interests> and add a statement declaring your competing interests. Please name that section 'Disclosure and Competing Interests Statement' and add it after the author contributions section.

12) Please order the sections like this using these names:

Title page - Abstract - Keywords - Introduction - Results - Discussion - Methods - Data availability section (DAS) - Acknowledgements (including funding information) - Disclosure and Competing Interests Statement - References - Figure legends - Expanded View Figure legends

13) Please provide the abstract written in present tense throughout.

14) Please make sure that all the funding information is also entered into the online submission system and is complete and similar to the one in the manuscript text file (in the Acknowledgements).

15) We now use CRediT to specify the contributions of each author in the journal submission system. CRediT replaces the author contribution section. Please use the free text box to provide more detailed descriptions. Thus, please do NOT provide your final manuscript text file with an author contributions section. See also guide to authors:

<https://www.embopress.org/page/journal/14693178/authorguide#authorshipguidelines>

16) All materials and methods used need to be described in the main text using our 'Structured Methods' format, which is required for all research articles. According to this format, the Methods section should include a Reagents and Tools Table (listing key reagents, experimental models, software, and relevant equipment and including their sources and relevant identifiers), uploaded as separate file, followed by a Methods section in which we encourage the authors to describe their methods using a step-by-step protocol format with bullet points, to facilitate the adoption of the methodologies across labs. More information on how to adhere to this format as well as downloadable templates (.doc or .xls) for the Reagents and Tools Table can be found in our author guidelines (section 'Structured Methods'):

I look forward to seeing a revised form of your manuscript when it is ready.

Yours sincerely,

Referee #1:

Following TCR activation, phospholipase CG1 (PLCG1) is recruited to the LAT signalosome where it hydrolyzes phosphatidylinositol 4,5- biphosphate (PIP2) to generate inositol trisphosphate and diacylglycerol that induce calcium influx and PKC activation, respectively. PLC γ 1 is one of the most frequently mutated gene in adult T-cell leukemia/lymphoma and the present study characterizes the functional impact of three of those mutations (R48W, S345F, and D1165H) in transformed cell lines. Using an assay that relies on total internal reflection fluorescence microscopy and monitors LAT condensation in presence of physiologically relevant concentrations of PLCG1, Grb2, and Sos1, they showed that the three PLCG1 mutants induced higher condensation of LAT as compared to the wild-type PLCG1. A similar conclusion was reached using Jurkat T cells lentivirally transduced with LAT-mCherry and wild-type or mutant PLCG. Two PLCG1 mutants (S345F and D1165H) increased

phosphorylation of PLCG1 and the three mutants showed higher ERK phosphorylation and calcium influx as compared to those expressing the wild-type PLCG1. Note that Jurkat cells kept their endogenous copy of PLCG1 which may account for the relatively modest effect that were observed for the two functional readouts. Similar modest increases were obtained in shortly activated primary human T cells. The authors next assessed the effect of the three mutants in Hut78, a human T lymphoma cell line. In absence of TCR engagement, they observed modestly enhanced phosphorylation of PLCG1 and ERK. It correlated with enhanced phosphorylation of STAT3 and HSP27, and reduced phosphorylation of Chk-2. A significant increase of ICAM-1 expression in Hut78 expressing PLCG1 mutants was also observed, resulting in aggregation of Hut78 cells. Moreover, it was found that the enhanced ERK signaling mediated by the PLCG1 mutant induced resistance to HDAC inhibitors. Bulk RNA sequencing of Hut78 cells expressing the wild-type or mutant PLCG1 and of untransfected Hut78 cells by anti-CD3/CD28 antibodies suggested a high expression of smooth muscle actin α -SMA, a marker of smooth muscle differentiation pathway, in Hut78 expressing PLCG1 mutants. Pull-down and SDS-PAGE profiling experiments were then performed using anti-EGFP antibodies and Hut78 cells expressing GFP-tagged wild-type or mutant PLCG1. A few specific bands appeared in the S345F and D1165H but not the WT samples. Their analysis by MS suggested that smooth muscle actin binds and stimulates the activation of PLCG1 leading to reinforce the hyperactive signaling in PLCG1 mutants. In conclusion, this descriptive study based on Hut78 cells suggest that some (see specific comment 6) hyperactive PLCG1 can promote T cell survival and drug resistance through inducing non-canonical signaling. It remains to be established whether this intriguing finding applies to freshly isolated adult T-cell leukemia/lymphoma samples. No mechanistic hints are also provided on the way alpha smooth muscle actin promotes PLCG1 its activation (see for instance 10.1006/bbr.1996.1735).

Specific comments:

- 1/ The authors should specify whether all the experiments reported in Figure 3 were done in absence of TCR stimulation.
- 2/ Figure E. For the sake of comparison, the authors should indicate the levels of IL-2 and TGF- β that will be produced following TCR/CD28 or PMA/iono stimulation.
- 3/ Page 10. The authors indicate "On the other hand, we determined the surface expression of common adhesion molecules and found a significant increase of ICAM-1 expression in Hut78 expressing PLCG1 mutants". Instead of 'significant increase' they should specify the fold change in MFI. The same applies to other parts of the Results.
- 4/ Page 12. I am missing the link between ACTG2 and smooth muscle actin α -SMA.
- 5/ Page 13. The logic behind the following paragraph is difficult to grasp: "These bands were cut out and sent for mass spectrometry analysis. They were identified as myosin heavy chain (around 220 kDa), myosin light chain (around 20 kDa) and actin (around 40 kDa) (Table S3). Because the pull-down of actin is specifically detected in D1165H mutant but not the wild-type PLCG1, and because the expression of alpha-SMA, but not that of beta actin or gamma actin was different between the wild-type and D1165H PLCG1 (Fig. 6E and 6F), we reasoned that alpha-SMA is likely to directly interact with PLCG1. Indeed, the recombinant PLCG1 D1165H protein binds to filamentous alpha-SMA protein in an actin co-pelleting assay (Fig. 7B and 7C). It is noted that the wild-type PLCG1 can also directly bind alpha-SMA, suggesting the binding to alpha-SMA is not restricted to mutant PLCG1'.
- 6/ The three PLCG1 mutants behave rather similarly in most of the functional assays. However, in the pull-down experiment shown in Figure 7 A and B, mutant R48W totally differed from the two other mutants. Have the authors an explanation this difference which question the main conclusion of their study? Along that line, it will have been useful to describe the PLCG1 structural domains that are affected by R48W, S345F, and D1165H.
- 7/ Have the authors analyse transcriptomics dataset corresponding to adult T-cell leukemia/lymphoma to document the presence of muscle actin?

Referee #2:

The authors investigate some characteristics of PLCG1 mutants associated with T cell leukaemia and lymphoma in leukaemia. They do this in a number of settings- from in vitro LAT condensates, in vivo LAT condensates and finally functional effect in a T lymphoma cell line. The study generates interesting, but somewhat divergent results related to LFA-1/ICAM-1 mediated adhesion and upregulation of smooth muscle actin, which may interact with PLCG1 to activate it. It seems like PLCG1 mutants are doing multiple things in conventional T cell signalling and some non-conventional pathways. I have some question suggestions. Is it possible to try to link the α SMA and LFA-1 components or are they completely parallel, although potentially hard to disentangle.

The LAT condensate work is clear. One of the authors has previously shown that F-actin (presumably β -actin) is incorporated into the LAT condensates on the SLB platform. Would it be possible to see if anything different happens when PLCG1 is driving LAT condensates and F actin made of conventional β -actin vs alpha smooth muscle actin is used instead. That might bring these two components of the paper together.

The authors mention that PLCG1 copies in Jurkat and Hut78 were left intact in cells transfected with the PLCG1 mutants. The authors should confirm that they have sequence for Jurkat and Hut78 PLCG1 (or found sequences in the literature) confirming that all copies are wild type. Jurkat was from a patient with an acute T cell leukemia and Hut78 a lymphoma as they point out. If they already have mutant copied then these may have undergone some selection for sustained growth and survival. That seems important to check.

The increased ICAM-1 expression may be a consequence of the interaction with LFA-1, trapping more ICAM-1 at the surface from an intracellular pool. Can the authors check total ICAM-1 levels by Western blotting and ICAM-1 mRNA levels by QPCR or other methods to determine if the increased surface expression is regulated at the level of protein distribution or mRNA? It's also not clear that the {greater than or equal to} 2-fold increase in ICAM-1 could even account for the increase in aggregation. It's more likely this is related to inside-out signalling through LFA-1. When cells expressing WT PLCG1 co-aggregate with cells expressing the mutated PLCG1, does the ICAM-1 on the WT PLCG1 expressing cells increase? ICAM-1 signalling to ERK is documented so it's possible that LFA-1 activation is driving ICAM-1 engagement leading to increased surface distribution and signalling. The result that the PLCG1 mutants induce resistance to vorinostat that is reversed by blocking LFA-1 dependent adhesion is interesting. Does this also require alpha smooth muscle actin (see below)?

The results seem to diverge quite a bit around some of the gene expression profiles in Hut78. It seems that the mutant PLCG1 take a cell that is already an immortalized tumour cell line that is well adapted to culture and generate additional changes in gene expression that are of unclear significance, but some plausible models are suggested and partly tested.

The expression levels of ITGA1 and ITGB5 are interesting, but other than being integrin subunits these seem unrelated. ITGA1 is part of a collagen binding integrin that is often expressed on tissue resident T cells, whereas ITGB5 is a partner of ITGA1 that is expressed in many cancer cells, but I'm not aware of a history of expression on T cells. It's also notable that SMAMF1 is increased, which is a homophilic adhesion molecule that might also contribute to homotypic T cell interactions. One problem with looking at relative expression levels is its good at assessing changes vs WT, but useless at assessing the overall level of expression and if the expression level predicts a likely function. It would be very straightforward for the authors to look at the surface expression of SLAMF1, ITGA1 and ITGB5 on the mutant PLCG1 expressing Hut78 to see what the surface levels are significant and the proteins warrant further study in this setting.

The story with smooth muscle associated cytoskeletal proteins is intriguing. There are suggestions in earlier papers that the PH domain and CSH2 may interact with F-actin generally. The authors don't really show any selectivity of the interaction with smooth muscle vs conventional beta-actin. Would these components be likely to form mixed filaments in cells and wouldn't it make sense to compare PLCG1 interaction with conventional F-actin filaments vs those containing alpha smooth muscle actin? I mentioned above the possibility of bringing this back into the LAT condensate story.

The results with latrunculin and jasplakinolide are hard to interpret. If the PLCG1 activation requires LFA-1 function then latrunculin should reduce LFA-1 function independent of PLCG1 binding to F-actin. ICAM-1 signaling would also likely require ICAM1. It's not as obvious to me that jas would necessarily result in better function of LFA-1 or ICAM-1 at least the form would require dynamic F-actin structures. ICAM-1 might be able to work well with more static F-actin. But this seems like a very crude experiment. Does expressing alpha smooth muscle actin in Hut78 recapitulate any of these results without needing the mutated PLCG1?

Minor

1. Page 5- LAF-1 to LFA-1

Referee #3:

Zeng et al investigate how expression of mutant forms of phospholipase C 1 (PLGC1) might lead to transformation of cells and, in particular, to development of T cell leukemias and lymphomas. Using standard in vitro assays, they demonstrate that the expression of three separate mutations leads to increased T cell signaling. Their work concludes with sections on how the mutations enhance resistance to a drug approved for treatment of a T cell malignancy and how resistance can be overcome based on the data from the initial part of the study. An additional mechanism of enhanced activation was shown to be binding of a form of smooth muscle actin to PLCG1, which directly activates the enzyme.

This is an excellent study that provides much novel information about potentially oncogenic mutations in PLCG1. I have some minor issues to raise, which can be easily addressed.

The effects of the three mutations as measured in various assays differ in intensity. Usually, though there are exceptions, the rank order of the mutations from most to least active is D1165H, S345F, R48W. I would have commented on those findings throughout the results section. I think the authors can directly relate and describe the findings to be the consequence of the enhanced specific activity of the mutants, i.e. the highest being D1165H. There is a brief mention of this observation and conclusion in the Discussion, but there I would have also addressed whether there is any clinical correlation to the rank-order finding. Such data might not be available, but that issue of potential clinical relevance could at least be mentioned.

There are several points where the writing could be corrected or clarified:

- The first sentence of the abstract is missing a verb (likely, "...in cell signaling field (is) to identify...") and an article, "...in (the) cell signaling field...."
- In the middle of the abstract add a hyphen to make text "leukemia/lymphoma-associated mutations"
- Clarify in the text and figure legends that the Sos1 preparation used in the study is a Grb2-binding fragment and not the whole protein. That becomes obvious in FigS1, but all the readers might not look at the supplementary figures. And regarding that figure, can the authors explain the several bands in the SOS1 lane?
- Clarify the text in the first paragraph in which Figure 3 is discussed. In the middle of the paragraph we learn that "...the expression levels of the mutants were lower than the wild-type...." but later in the paragraph we learn "...that the expression of these...mutants...was much higher than that of endogenous PLCG1"

EMBOR-2025-61249-T

Response Letter

This manuscript was originally submitted to EMBO Journal and sent out for review. Based on the reviewers' comments (detailed below), it was recommended to be transferred to EMBO Reports. Below is the editorial guidance from EMBO Reports:

EMBO Reports emphasizes novel functional insight with clear in vivo relevance over detailed mechanistic insight. Thus, EMBO Reports will not require addressing points regarding more mechanism experimentally, but welcomes such data, in case you already have it. However, it will be necessary that during the revision you address all points questioning the main conclusions of the study, and all technical concerns, or points regarding the experimental designs, model systems used, or data presentation.

In this case, EMBO reports does not need data from freshly isolated adult T-cell leukemia/lymphoma samples (referee #1), and - as already mentioned - does not require further mechanistic insight. However, other referee points that refer to the current conclusions of the study need attention, in particular the second point of referee #2 in order to show that the endogenous copies of Plcg1 in the used cell lines are indeed wild type.

We appreciate this editorial guidance and the waiver of experiments involving patient samples and further mechanistic studies. We will focus on addressing all the other reviewers' concerns.

Referee #1:

Following TCR activation, phospholipase CG1 (PLCG1) is recruited to the LAT signalosome where it hydrolyzes phosphatidylinositol 4,5- bisphosphate (PIP2) to generate inositol trisphosphate and diacylglycerol that induce calcium influx and PKC activation, respectively. PLCγ1 is one of the most frequently mutated gene in adult T-cell leukemia/lymphoma and the present study characterizes the functional impact of three of those mutations (R48W, S345F, and D1165H) in transformed cell lines. Using an assay that relies on total internal reflection fluorescence microscopy and monitors LAT condensation in presence of physiologically relevant concentrations of PLCG1, Grb2, and Sos1, they showed that the three PLCG1 mutants induced higher condensation of

LAT as compared to the wild-type PLCG1. A similar conclusion was reached using Jurkat T cells lentivirally transduced with LAT-mCherry and wild-type or mutant PLCG. Two PLCG1 mutants (S345F and D1165H) increased phosphorylation of PLCG1 and the three mutants showed higher ERK phosphorylation and calcium influx as compared to those expressing the wild-type PLCG1. Note that Jurkat cells kept their endogenous copy of PLCG1 which may account for the relatively modest effect that were observed for the two functional readouts. Similar modest increases were obtained in shortly activated primary human T cells. The authors next assessed the effect of the three mutants in Hut78, a human T lymphoma cell line. In absence of TCR engagement, they observed modestly enhanced phosphorylation of-PLCG1 and-ERK. It correlated with enhanced phosphorylation of STAT3 and HSP27, and reduced phosphorylation of Chk-2. A significant increase of ICAM-1 expression in Hut78 expressing PLCG1 mutants was also observed, resulting in aggregation of Hut78 cells. Moreover, it was found that the enhanced ERK signaling mediated by the PLCG1 mutant induced resistance to HDAC inhibitors. Bulk RNA sequencing of Hut78 cells expressing the wild-type or mutant PLCG1 and of untransfected Hut78 cells by anti-CD3/CD28 antibodies suggested a high expression of smooth muscle actin α -SMA, a marker of smooth muscle differentiation pathway, in Hut78 expressing PLCG1 mutants. Pull-down and SDS-PAGE profiling experiments were then performed using anti-EGFP antibodies and Hut78 cells expressing GFP-tagged wild-type or mutant PLCG1. A few specific bands appeared in the S345F and D1165H but not the WT samples. Their analysis by MS suggested that smooth muscle actin binds and stimulates the activation of PLCG1 leading to reinforce the hyperactive signaling in PLCG1 mutants. In conclusion, this descriptive study based on Hut78 cells suggest that some (see specific comment 6) hyperactive PLCG1 can promote T cell survival and drug resistance through inducing non-canonical signaling. It remains to be established whether this intriguing finding applies to freshly isolated adult T-cell leukemia/lymphoma samples. No mechanistic hints are also provided on the way alpha smooth muscle actin promotes PLCG1 its activation (see for instance 10.1006/bbrc.1996.1735).

We thank the reviewer for insightful and constructive comments. The editorial from EMBO Reports waived the experiments on freshly isolated T cell patient samples and more mechanistic studies. We focused on addressing the rest points as below.

Specific comments:

1/ The authors should specify whether all the experiments reported in Figure 3 were done in absence of TCR stimulation.

This is a great point. We now specified in both the main text and figure legend that all experiments reported in Figure 3 were done in the absence of TCR stimulation.

2/ Figure E. For the sake of comparison, the authors should indicate the levels of IL-2 and TGF- β that will be produced following TCR/CD28 or PMA/iono stimulation.

We measured the secretion of IL-2 and TGF- β following TCR/CD28 stimulation (Fig 3E). We found that all the three mutants showed an over 10-fold increase in IL2 and an over 6-fold increase in TGF- β production as compared to the WT.

(Fig. 3E) ELISA analysis of IL2 and TGF- β secretion by Hut78 cells in resting versus activation states. Hut78 cells were activated by anti-CD3/CD28 antibodies for 72 hours. Shown are mean \pm SD from n=3 biological replicates. Unpaired two-tailed t-test was used.

3/ Page 10. The authors indicate "On the other hand, we determined the surface expression of common adhesion molecules and found a significant increase of ICAM-1 expression in Hut78 expressing PLCG1 mutants". Instead of 'significant increase' they should specify the fold change in MFI. The same applies to other parts of the Results.

We specified the fold change throughout the manuscript.

4/ Page 12. I am missing the link between ACTG2 and smooth muscle actin α -SMA.

Both ACTG2 and α -SMA are in the same transcriptional program to define the pathway of smooth muscle contraction and both are upregulated in PLCG1 mutants. We mentioned this in the text now.

5/ Page 13. The logic behind the following paragraph is difficult to grasp: "These bands were cut out and sent for mass spectrometry analysis. They were identified as myosin heavy chain (around 220 kDa), myosin light chain (around 20 kDa) and actin (around 40 kDa) (Table S3). Because the pull-down of actin is specifically detected in D1165H mutant but not the wild-type PLCG1, and because the expression of alpha-SMA, but not that of beta actin or gamma actin was different between the wild-type and D1165H

PLCG1 (Fig. 6E and 6F), we reasoned that alpha-SMA is likely to directly interact with PLCG1. Indeed, the recombinant PLCG1 D1165H protein binds to filamentous alpha-SMA protein in an actin co-pelleting assay (Fig. 7B and 7C). It is noted that the wild-type PLCG1 can also directly bind alpha-SMA, suggesting the binding to alpha-SMA is not restricted to mutant PLCG1'.

We apologize for the difficulty in understanding the logic of this paragraph. We rephrased it in the text to provide a clear rationale on the actin-binding assay.

6/ The three PLCG1 mutants behave rather similarly in most of the functional assays. However, in the pull-down experiment shown in Figure 7 A and B, mutant R48W totally differed from the two other mutants. Have the authors an explanation for this difference which question the main conclusion of their study? Along that line, it will have been useful to describe the PLCG1 structural domains that are affected by R48W, S345F, and D1165H.

A previous report showed that among the three mutants, D1165H showed the highest enzymatic activity, which is followed by S345F, whereas R48W is the lowest (PMID: 31889510). We observed a similar trend in our signaling assay when measuring the level of phospho-PLCG1 and phospho-ERK (Fig. 3A). Therefore, these results are consistent with the data that R48W did not pull down actin in Figure 7. We agree with the reviewer that this difference between mutants might be related to the structural domain where these mutations are located. R48W is in the N-terminal PH domain that potentially interacts with the plasma membrane. This is far away from the catalytic core. In contrast, both S345F and D1165H are located in the interface between the regulatory cluster and catalytic core. These two mutations are expected to favor an open conformation of PLCG1 for hydrolyzing PIP2 (PMID: 31889510). We will emphasize these in the introduction part.

7/ Have the authors analyse transcriptomics dataset corresponding to adult T-cell leukemia/lymphoma to document the presence of muscle actin?

This is a great idea. We analyzed the transcriptomics dataset of adult T-cell leukemia/lymphoma from published datasets and found that the alpha-SMA expression is significantly upregulated in patients compared to healthy donors (Fig. 8D).

D

(Fig 8D) RNA-seq analysis of adult T-cell leukemia/lymphoma samples. The expressions of Alpha smooth muscle actin (*ACTA2*) and ICAM-1 (*ICAM1*) in ATLL samples (n=66) were compared to those in healthy donors (n=3). Unpaired two-tailed Welch's t-test was used.

Referee #2:

The authors investigate some characteristics of PLCG1 mutants associated with T cell leukaemia and lymphoma in leukaemia. They do this in a number of settings- from in vitro LAT condensates, in vivo LAT condensates and finally functional effect in a T lymphoma cell line. The study generates interesting, but somewhat divergent results related to LFA-1/ICAM-1 mediated adhesion and upregulation of smooth muscle actin, which may interact with PLCG1 to activate it. It seems like PLCG1 mutants are doing multiple things in conventional T cell signalling and some non-conventional pathways. I have some question suggestions. Is it possible to try to link the aSMA and LFA-1 components or are they completely parallel, although potentially hard to disentangle.

We thank the reviewer for raising the idea of integrating aSMA and LFA-1 part. Meanwhile, in our current model, α -SMA and LFA-1 function in parallel although they converge on ERK activation. α -SMA binds PLCG1 and activates it, which leads to the activation of ERK through conventional TCR pathway. LFA-1 is activated by ICAM-1 on neighboring cells, which leads to ERK activation as well. Both pathways enhance ERK activation in parallel, which accounts for drug resistance to HDAC inhibitors. However, we confess, as the reviewer pointed out, that it is difficult to mechanistically determine whether aSMA and LFA-1 pathways are completely parallel or linked to some extent.

The LAT condensate work is clear. One of the authors has previously shown that F-actin (presumably β -actin) is incorporated into the LAT condensates on the SLB platform. Would it be possible to see if anything different happens when PLCG1 is driving LAT condensates and F actin made of conventional β -actin vs α smooth muscle actin is used instead. That might bring these two components of the paper together.

We thank the reviewer for recognizing Dr. Su's previous work showing how LAT condensates drive β -actin polymerization in vitro. This could be an interesting assay to reveal more **mechanistic insights** on how LAT condensates, as influenced by PLCG1 mutants, regulate actin polymerization. However, because Dr. Su performed this assay about 10 years ago when he was a postdoc in Ron Vale's Lab. The ~20 proteins used in the assay were collected from three labs (Ron Vale, Mike Rosen, and Jack Tauron). The technical barrier on performing this assay is stunningly high. Moreover, the pathway that links LAT to actin polymerization has been well established. The conduct of this assay is not likely to reveal any new mechanism. Therefore, we ask permission to waive this experiment.

The authors mention that PLCG1 copies in Jurkat and Hut78 were left intact in cells transfected with the PLCG1 mutants. The authors should confirm that they have sequence for Jurkat and Hut78 PLCG1 (or found sequences in the literature) confirming that all copies are wild type. Jurkat was from a patient with an acute T cell leukemia and Hut78 a lymphoma as they point out. If they already have mutant copied then these may have undergone some selection for sustained growth and survival. That seems important to check.

We agree with the reviewer that it is critical to determine the sequence of PLCG1 in Jurkat and Hut78. We performed whole genome sequencing analysis in both cells (Appendix Fig. S1) and found that 1) The three ATLL mutations examined in this study R48W, S345F, and D1165H, are **absent** in Jurkat and Hut78; 2) Jurkat harbors a common variant (defined by >5% in human population) at PLCG1-T2438C, which has a frequency of 26.7%; 3) Hut78 harbors two common variants: one is PLCG1-T2438C and the other is A835G, which has a frequency of 7.9%; 4) The two variants cause a change in amino acid sequence (I813T and S279G) as compared to the reference sequence. Neither mutation is present in ATLL patients (PMID:26437031); 5) Based on the crystal structure of PLCG1 (PMID: 31889510), I813T and S279G are located far away from the catalytic core and the regulatory domain interface, which is in contrast with the ATLL mutants (Figure to reviewer 1 - *redacted*). The residue at 813 and 279 did not form chemical bonds with other residues. Using alphafold III prediction, I813T and S279G did not induce a significant change in local or global structure. These observations suggest that I813T and S279G are not likely to alter the structure and activity of PLCG1.

Meanwhile, we understand that it is difficult to exclude the minor effect that these two variants could have on PLCG1 function. Moreover, the cancer cell lines used in this study might contain other mutations that affect the signaling outcomes. Therefore, we confirmed the major findings of this study using human primary T cells. We showed that the ATLL mutants cause higher cell proliferation and survival (Fig. 2G, 2H), ERK phosphorylation (Fig. 8A), and induce the expression of ICAM1 and aSMA (Fig. 8A-8C).

Appendix Fig S1

A

Cell line	Gene variant	Codon Change	Mutation type	Amino acid mutation	Minor allele frequencies-1000 genomes projects	dbSNP ID
Jurkat	PLCG1-T2438C	aTc/aCc	T(0%)/C(100%), Homozygous	PLCG1-I813T	26.7%, Common variant (>5%)	rs753381
Hut78	PLCG1-A835G	Agc/Ggc	A(43%)/G(57%), Heterozygous	PLCG1-S279G	7.9%, Common variant (>5%)	rs2228246
Hut78	PLCG1-T2438C	aTc/aCc	T(43%)/C(57%), Heterozygous	PLCG1-I813T	26.7%, Common variant (>5%)	rs753381

B

C

D

E

Appendix for Figure S1 Legend:

PLCG1 sequence in Jurkat and Hut78 cells as revealed by whole-genome sequencing.

- (A) Summary of common variants in PLCG1 in Jurkat and Hut78 cells
- (B) Local view of the PLCG1 common variants in Jurkat and Hut78 cells as compared to reference human genome sequence.
- (C) Local view of sequences encoding R48.
- (D) Local view of sequences encoding S345.
- (E) Local view of sequences encoding D1165.

Figure for referee with unpublished data and its description has been removed upon request by the authors.

The increased ICAM-1 expression may be a consequence of the interaction with LFA-1, trapping more ICAM-1 at the surface from an intracellular pool. Can the authors check total ICAM-1 levels by Western blotting and ICAM-1 mRNA levels by QPCR or other methods to determine if the increased surface expression is regulated at the level of protein distribution or mRNA? It's also not clear that the {greater than or equal to} 2-fold increase in ICAM-1 could even account for the increase in aggregation. It's more likely this is related to inside-out signalling through LFA-1. When cells expressing WT PLCG1 co-aggregate with cells expressing the mutated PLCG1, does the ICAM-1 on the WT PLCG1 expressing cells increase? ICAM-1 signalling to ERK is documented so it's possible that LFA-1 activation is driving ICAM-1 engagement leading to increased surface distribution and signalling. The result that the PLCG1 mutants induce resistance to vorinostat that is reversed by blocking LFA-1 dependent adhesion is interesting. Does this also require alpha smooth muscle actin (see below)?

We thank the reviewer for these insightful suggestions. We determined the mRNA level of ICAM1 by qPCR and total protein level by western blotting. Consistent with our previous flow data (Fig. 4B-4C), we found that ICAM1 is upregulated both at the RNA and protein level in PLCG1 mutants (Fig 4D-4F).

(Fig 4D) The ICAM-1 mRNA level as determined by qPCR. Shown are mean \pm SD from $n=3$ biological replicates. Unpaired two-tailed t-test was used.

(Fig 4E) The ICAM-1 protein level as determined by western blot.

(Fig 4F) Quantification of (E). Shown are mean \pm SD from $n=3$ biological replicates. Unpaired two-tailed t-test was used.

We measured the surface expression of ICAM-1 on the WT PLCG1-expressing cells when they were mixed with either the WT or the mutant-expressing cells. The data showed that the surface ICAM-1 expression on the WT PLCG1-expressing cells is increased by the mutant-expressing cells (Fig EV3E and EV3F). This supports the reviewer's hypothesis that LFA-1 activation is driving ICAM-1 engagement leading to increased surface distribution and signalling.

(Fig EV3E) Schematics of the co-culture assay. Violet-labeled Hut78 cells were co-cultured with Hut78 cells expressing PLCG1-GFP WT or D1165H in a 1:1 ratio for 72 h.

(Fig EV3F) ICAM-1 cell surface expression by FACS. Shown are mean \pm SD from $n=3$ biological replicates. Unpaired two-tailed t-test was used.

Quoting the reviewer “The result that the PLCG1 mutants induce resistant to vorinostat that is reversed by blocking LFA-1 dependent adhesion is interesting”. We did not have this data in our manuscript. However, this result might be expected based on our data that the PLCG1 mutants induce pERK, which is reversed by blocking LFA-1.

The results seem to diverge quite a bit around some of the gene expression profiles in Hut78. It seems that the mutant PLCG1 take a cell that is already an immortalized tumour cell lines that is well adapted to culture and generate additional changes in gene expression that are of unclear significance, but some plausible models are suggested and partly tested.

The expression levels of ITGA1 and ITGB5 are interesting, but other than being integrin subunits these seem unrelated. ITGA1 is part of a collagen binding integrin that is often expressed on tissue resident T cells, whereas ITGB5 is a partner of ITGAV that is expressed in many cancer cells, but I'm not aware of a history of expression on T cells. It's also notable that SLAMF1 is increased, which is a homophilic adhesion molecule that might also contribute to homotypic T cell interactions. One problem with looking at relative expression levels is its good at assessing changes vs WT, but useless at assessing the overall level of expression and if the expression level predicts a likely function. It would very straight forward for the authors to look at the surface expression of SLAMF1, ITGA1 and ITGB5 on the mutant PLCG1 expressing Hut78 to see what the surface levels are significant and the proteins warrant further study in this setting.

As suggested by the reviewer, we compared the surface expression of SLAMF1, ITGA1 and ITGB5 between the wild-type and mutant-expressing cells (Fig EV5B). We found that PLCG1 mutations increase the surface expression of CD49a (ITGA1) by 2- to 2.4-fold, CD150 (SLAMF1) by 1.6-3.1-fold, and integrin β 5 (ITGB5) by 1.6-3.3 fold.

(Fig EV5B) Cell surface protein expression by flow cytometry. Shown are mean \pm SD from $n=3$ biological replicates. Unpaired two-tailed t-test was used.

The story with smooth muscle associated cytoskeletal proteins is intriguing. There are suggestions in earlier papers that the PH domain and CSH2 may interact with F-actin generally. The authors don't really show any selectivity of the interaction with smooth muscle vs conventional beta-actin. Would these components be likely to form mixed filaments in cells and wouldn't it make sense to compare PLCG1 interaction with conventional F-actin filaments vs those containing alpha smooth muscle actin? I mentioned above the possibility of bringing this back into the LAT condensate story.

We compared the binding of PLCG1 to beta-actin filament versus α -SMA filament. The actin-binding assay indicates that F- α -SMA filaments exhibit a PLCG1-binding capacity comparable to that of conventional F-actin (Figure to Reviewer 2 - *redacted*). We think there might be additional factors regulating the interaction between α -SMA and PLCG1 in cells, which could be interesting to explore in the future.

Figure for referee with unpublished data and its description has been removed upon request by the authors.

The results with latrunculin and jasplakinolide are hard to interpret. If the PLCG1 activation requires LFA-1 function then latrunculin should reduce LFA-1 function independent of PLCG1 binding to F-actin. ICAM-1 signaling would also likely require ICAM1. It's not as obvious to me that jas would necessarily result in better function of LFA-1 or ICAM-1 at least the form would require dynamic F-actin structures. ICAM-1 might be able to work with more static F-actin. But this seems like a very crude experiment. Does expressing alpha smooth muscle actin in Hut78 recapitulate any of these results without needing the mutated PLCG1?

Based on our current model, we don't think PLCG1 activation requires LFA-1 function. This is supported by the data below showing that LFA-1 blocking did not affect PLCG1 phosphorylation (Figure to reviewer 3A - *redacted*). In addition, we determined if ectopically expressing α -SMA in Hut78 is sufficient to enhance pERK and ICAM1, the key signaling consequences of PLCG1 mutants. However, we did not observe that (Figure to Reviewer 3B - *redacted*), suggesting α -SMA is necessary but not sufficient to drive PLCG1 signaling. Some other cofactors might be required in this process.

Figure for referee with unpublished data and its description has been removed upon request by the authors.

In addition to pharmacological disruption of actin filaments using latrunculin and jasplakinolide, we performed α -smooth muscle actin (α -SMA) knockout in Hut78 cells and found that α -SMA KO reduces PLCG1 phosphorylation (Fig. 7D and 7E). This provides genetic evidence, in addition to the pharmacological evidence we provided, to support the conclusion that α -SMA promotes PLCG1 activation.

(Fig. 7D) Immunoblotting of PLCG1 phosphorylation in alpha SMA-knockout Hut78 cells.

(Fig. 7E) Quantification of PLCG1 phosphorylation from (D). Shown are mean \pm SD from n=3 biological replicates. Unpaired two-tailed t-test was used.

Minor

1. Page 5- LAF-1 to LFA-1

Eagle eye! Typo corrected.

Referee #3:

Zeng et al investigate how expression of mutant forms of phospholipase C γ 1 (PLGC1) might lead to transformation of cells and, in particular, to development of T cell leukemias and lymphomas. Using standard in vitro assays, they demonstrate that the expression of three separate mutations leads to increased T cell signaling. Their work concludes with sections on how the mutations enhance resistance to a drug approved for treatment of a T cell malignancy and how resistance can be overcome based on the data from the initial part of the study. An additional mechanism of enhanced activation was shown to be binding of a form of smooth muscle actin to PLGC1, which directly activates the enzyme.

This is an excellent study that provides much novel information about potentially oncogenic mutations in PLGC1. I have some minor issues to raise, which can be easily addressed.

We thank the reviewer for high enthusiasm for our study.

The effects of the three mutations as measured in various assays differ in intensity. Usually, though there are exceptions, the rank order of the mutations from most to least active is D1165H, S345F, R48W. I would have commented on those findings throughout the results section. I think the authors can directly relate and describe the findings to be the consequence of the enhanced specific activity of the mutants, i.e. the highest being D1165H. There is a brief mention of this observation and conclusion in the Discussion, but there I would have also addressed whether there is any clinical correlation to the rank-order finding. Such data might not be available, but that issue of potential clinical relevance could at least be mentioned.

Great suggestions! We modified the discussion part accordingly.

There are several points where the writing could be corrected or clarified:

- The first sentence of the abstract is missing a verb (likely, "...in cell signaling field (is) to identify...") and an article, "...in (the) cell signaling field...."
- In the middle of the abstract add a hyphen to make text "leukemia/lymphoma-associated mutations"

We corrected these typos.

- Clarify in the text and figure legends that the Sos1 preparation used in the study is a Grb2-binding fragment and not the whole protein. That becomes obvious in FigS1, but

all the readers might not look at the supplementary figures. And regarding that figure, can the authors explain the several bands in the SOS1 lane?

We now clarify that Sos1 used in this study is a fragment of proline-rich motifs. We used affinity purification followed by gel filtration to purify Sos1. The purity of the product is about 80-90%. We do notice a band of lower molecular weight in the Sos1 lane which could be a degradation product.

- Clarify the text in the first paragraph in which Figure 3 is discussed. In the middle of the paragraph we learn that "...the expression levels of the mutants were lower than the wild-type...." but later in the paragraph we learn ...that the expression of these...mutants...was much higher than that of endogenous PLCG1"

We apologize for this confusion. In the first sentence, both the wild-type and mutant PLCG1 were **ectopically expressed** in Hut78 cells. In the second sentence, the expression of the ectopically expressed mutants was much higher than that of **the endogenous level** of PLCG1. We now clarified this in the text.

Dear Dr. Su,

Thank you for the submission of your revised manuscript to our editorial offices. I have now received the reports from the three referees that I asked to re-evaluate the study, you will find below. As you will see, the referees now support the publication of your study.

Before we can proceed with formal acceptance, I have these editorial requests I ask you to address in a final revised manuscript:

- Please provide the abstract written in present tense throughout.
 - Please also provide the subtitles of the results part and the figure titles written in present tense.
 - We now use CRediT to specify the contributions of each author in the journal submission system. CRediT replaces the author contribution section. Please use the free text box to provide more detailed descriptions and do NOT provide your final manuscript text file with an author contributions section. See also our guide to authors: <https://www.embopress.org/page/journal/14693178/authorguide#authorshippinguidelines>
 - Please name the reference list 'References' and follow our journal style (use 'et al' if there are more than 10 author names). See: <http://www.embopress.org/page/journal/14693178/authorguide#referencesformat>
 - Please remove the mention of published datasets from the data availability section. This section is restricted to deposited datasets produced during the study.
 - Please add Kataoke 2015 and Kogure 2022 as data citations to the reference list and use appropriate callouts. See: <https://www.embopress.org/page/journal/14693178/authorguide#referencesformat>
 - Please add scale bars of similar style and thickness to microscopic images, using clearly visible black or white bars (depending on the background). Please place these in the lower right corner of the images themselves. Please do not write on or near the bars in the image but define the size in the respective figure legend. Presently, some scale bars are too small (e.g. in Fig. 1C), or have text nearby (e.g. in Fig. 4A). Please check.
 - I would suggest to move what is shown now in Appendix Fig. S1 to the EV figures. Then, no Appendix is needed. Please do that and update all the callouts.
 - Please check again that the number "n" for how many independent experiments were performed, their nature (biological versus technical replicates), the bars and error bars (e.g. SEM, SD) and the test used to calculate p-values is indicated in the respective figure legends. Please also check that all the p-values are explained in the legend, and that these fit to those shown in the figure. Please provide statistical testing where applicable. Please avoid the phrase 'independent experiment' but clearly state if these were biological or technical replicates. Please also indicate (e.g. with n.s.) if testing was performed, but the differences are not significant. In case n=2, please show the data as separate datapoints without error bars and statistics. See also: <http://www.embopress.org/page/journal/14693178/authorguide#statisticalanalysis>
- If n<5, please show single datapoints for diagrams. Some diagrams show only partial, or no statistics (Panel 4H) or the 'n.s.' seems missing. Please check. Moreover:
- Please note that the exact p values are not provided in the legends of figures 3D, E; 4C, 8C, EV4 C, EV5 C.
 - Please note that information related to n is missing in the legends of figures 5A, EV4 A, B.
 - Please note that the error bars are not defined in the legends of figures 5A, EV4 A, B
- Please make sure that all the funding information is also entered into the online submission system and that it is complete and similar to the one in the acknowledgement section of the manuscript text file. The comments box should not be used, only the separate funders via the 'More Funders' option; the following funders are only listed in the box, and need to be provided as separate funders: the Yale Cancer Center Pilot Award, the Yale DeLuca Pilot Award, the Gabrielle's Angel Foundation Medical Research Award, the Pershing Square Sohn Prize for Young Investigators in Cancer research, the Yale Lion Heart Pilot Grant, the Leslie Warner Postdoctoral Fellowship, the Daiichi Sankyo Foundation of Life Science Fellowship.
 - Tables EV1 and EV2 are datasets. Please name and upload these as Dataset EV1 and Dataset EV2 in all places (source file names, titles in the submission system and their callouts). Please remove their legends removed from the manuscript text file. Instead, please put these on the first TAB of the corresponding Excel file.
 - Please upload the Reagents & Tools table as separate file and remove it from the manuscript text. Please also add callouts to

the table where appropriate.

- Thank you for providing the requested source data (SD). Please upload this as one folder per main figure (with all files for one figure in one folder and ZIPed together).

In addition, I would need from you uploaded separately:

Referee #1:

I reviewed the revised manuscript and all the points I raised have been fully and fairly addressed. The same appears to apply to the points raised by the two other reviewers. In my view the MS is acceptable for publication as such.

Referee #2:

I was the most favorable of referees for the first draft, and would have accepted the paper with minor revisions. The authors have satisfied my requests and I think the paper is ready for publication.

Referee #3:

The authors have completely addressed the concern regarding ICAM-1 upregulation at mRNA and protein level. The authors have not addressed the implications of SMA for LFA-1 function (potential for increase contractile force/anchorage) or implication of higher SMA for interaction with LAT condensates as beyond scope. Given the amount of work done and the importance of the drug resistance I withdraw the request for these experiments.

All editorial and formatting issues were resolved by the authors.

Dr. Xiaolei Su
Yale School of Medicine
United States

Dear Dr. Su,

I am very pleased to accept your manuscript for publication in the next available issue of EMBO reports. Thank you for your contribution to our journal.

Yours sincerely,
